# Modeling Others' Minds as Code

**Kunal Jha[1], Aydan Yuenan Huang[2], Eric Ye[1],**
**Natasha Jaques[1]\* & Max Kleiman-Weiner[1]\***

## Abstract

Accurate prediction of human behavior is essential for robust and safe human-AI collaboration. However, existing approaches for modeling people are often data-hungry and brittle because they either make unrealistic assumptions about rationality or are too computationally demanding to adapt rapidly. Our key insight is that many everyday social interactions may follow predictable patterns; efficient "scripts" that minimize cognitive load for actors and observers, e.g., "wait for the green light, then go." We propose modeling these routines as behavioral programs instantiated in computer code rather than policies conditioned on beliefs and desires. We introduce ROTE, a novel algorithm that leverages both large language models (LLMs) for synthesizing a hypothesis space of behavioral programs, and probabilistic inference for reasoning about uncertainty over that space. We test ROTE in a suite of gridworld tasks and a large-scale embodied household simulator. ROTE predicts human and AI behaviors from sparse observations, outperforming competitive baselines—including behavior cloning and LLM-based methods—by as much as 50% in terms of in-sample accuracy and out-of-sample generalization. By treating action understanding as a program synthesis problem, ROTE opens a path for AI systems to efficiently and effectively predict human behavior in the real-world.

## 1 Introduction

Predicting the behavior of others (Theory of Mind) is a core challenge for building intelligent social agents. Whether anticipating a pedestrian's movements, coordinating with teammates, or interacting safely in public spaces, machines must infer what others are likely to do next. Existing approaches such as behavior cloning (BC) and inverse reinforcement learning (IRL) rely on learning models to predict low-level actions or infer latent reward functions (Abbeel & Ng, 2004; Ng et al., 2000; Torabi et al., 2018; Wulfmeier et al., 2016). However, these methods are often data-hungry and brittle because they try to learn what an agent might do in *every* possible state, frequently overfitting to specific environments or overcomplicating behaviors that are surprisingly routine for humans (Skalse & Abate, 2024; Yildirim et al., 2024). Alternatively, probabilistic methods for goal inference (Fuchs et al., 2023; Zhi-Xuan et al., 2020; 2024) are more sample efficient but demand computationally intensive online reasoning about potential intentions and beliefs, alongside human-specified priors and hypothesis spaces. Thus, conventional methods for modeling others present a trade-off illustrated in Figure 1: data-intensive and brittle, or compute-intensive and manually constructed for each new domain.

Recent work in cognitive science shows that when humans interact with one another, we do not always imbue others with deeply held mental states such as goals or beliefs. Instead we often perceive others as following a script or mindlessly applying a set of rules (Ullman & Bass, 2024; Bass et al., 2024). For example, when someone steps into a crosswalk, we do not need to infer their ultimate destination, their complex mental states, or their opinion on pineapple on pizza. It is enough to apply a commonly understood "crosswalk script" shaped by social convention. While there are perspectives on how people adopt roles in societies or prescribe agency to others (Dennett, 1972; Field, 1978; Dennett & Gorey, 1981; Dennett,

---

\*Equal contribution. [1] Department of Computer Science, University of Washington, Seattle, WA
[2] Department of Computer Science, Johns Hopkins University, Baltimore, MD

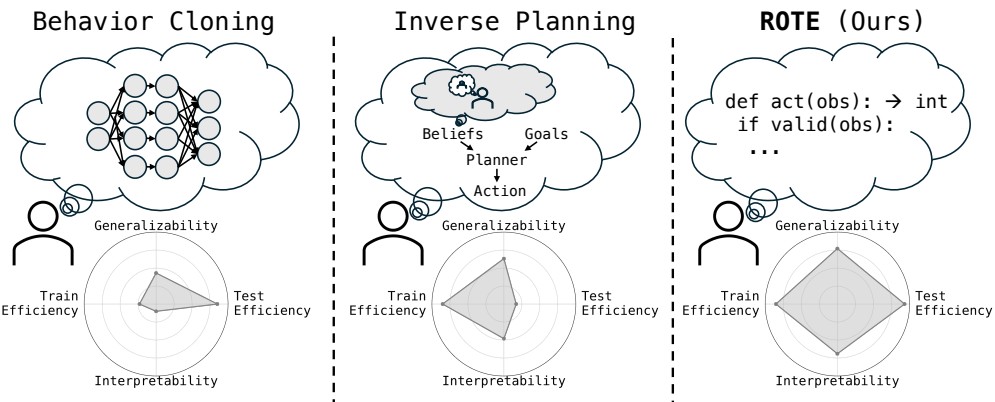

Figure 1: Comparison of action prediction methods: Behavior cloning requires large datasets and has limited generalization, while inverse planning is computationally expensive at test time. Our approach, ROTE, uses LLMs to generate efficient and interpretable code representations of observed behavior, providing a superior balance of efficiency and accuracy.

1987; 2017; Jara-Ettinger & Dunham, 2024), to the best of our knowledge, there are currently no computational models that adequately describe how machines can represent and reason about other agents acting in a script-like manner.

The notion of representing an intelligent agent through logical rules and predetermined decision-making processes is a foundational idea in computer science (Newell & Simon, 1956; Schank & Abelson, 2013; Newell & Simon, 1976), influencing fields from planning (Campbell et al., 2002; Zhu et al., 2025) to game theory (Axelrod, 1980). Finite State Machines (FSMs), for instance, are still used in video games to efficiently simulate large numbers of agents. By defining a sequence of states and transitions (e.g., patrol border → find agents → chase agents), code can flexibly model the causal behaviors underpinning social norms and routines.

Here we develop **ROTE** — **R**epresenting **O**thers' **T**rajectories as **E**xecutables — a novel algorithm that leverages LLMs as code synthesis tools to predict others' actions. We prompt LLMs to generate computer programs explaining observed behavioral traces, then perform Bayesian inference to reason about which programs are most likely. This gives us a dynamic representation that can be analyzed, modified, and composed across agents and environments.

ROTE significantly improves generalization and efficiency in predicting complex agent behavior, showing up to a 50% increase in accuracy across multiple challenging embodied domains. Our results in gridworlds and the scaled-up *Partnr* household robotics simulator demonstrate that code is a highly effective representation for modeling and predicting behavior. To validate its applicability to real-world complexity, we collected human gameplay data and found that *our method achieves human-level accuracy in predicting human actions*, outperforming all baselines. This offers a promising new path for creating scalable, adaptable, and interpretable socially intelligent AI systems. Concretely, our contributions are:

1. **Modeling Agentic Behavior via Program Synthesis:** We develop ROTE, a novel algorithm that combines LLMs with Sequential Monte Carlo to model other agents' behavior as programs from sparse observations.

2. **Superior, Scalable Action Prediction:** Across two embodied domains, we show that ROTE offers superior generalization for predicting others' behaviors, outperforming alternative methods by as much as 50%. Our method generates executable code that is reusable across environments, bypassing costly reasoning over goals and beliefs. These code-based representations scale more efficiently than behavior cloning or inverse planning alternatives, even when the ground truth behavior does not come from a known program.

3. **Human Studies Validation:** We recruit real human participants to generate behavior and predict others' actions. We find that ROTE outperforms baselines and *achieves human-level performance in predicting human behaviors*, even for noisy and sparse trajectories.

## 2 RELATED WORK

**Action Prediction.** Prior work developing AI for action prediction follows two dominant categories: symbolic methods and neural networks. Symbolic methods, such as Bayesian Inverse Planning (BIP), infer an agent's goals and beliefs by calculating their probabilities based on observed actions (Ullman et al., 2009; Baker et al., 2017; Shum et al., 2019; Netanyahu et al., 2021; Kleiman-Weiner et al., 2016; Wang et al., 2020; Kleiman-Weiner et al., 2020; Serrino et al., 2019; Kleiman-Weiner et al., 2025). While robust, these methods are not scalable due to the exponential complexity of a multi-agent environment (Rathnasabapathy et al., 2006; Doshi & Gmytrasiewicz, 2009; Seaman et al., 2018). In contrast, neural approaches like behavioral cloning (BC) and inverse reinforcement learning (IRL) train models to directly mimic actions (Torabi et al., 2018; Ng et al., 2000; Abbeel & Ng, 2004; Wulfmeier et al., 2016; Wang et al., 2021; Christiano et al., 2023), but are often data-intensive, fragile, and prone to overfitting. Recent work has tried modeling reward functions as finite-state automatons, a concept known as "reward machines" (Icarte et al., 2018; Toro Icarte et al., 2022; Li et al., 2025). This method, which does not use LLMs, allows for structured representation of reward and can provide non-Markovian feedback to agents. While primarily used for training agents to solve compositional tasks, there has been work on inferring reward machines from expert demonstrations (Zhou & Li, 2022) or learning safety constraints (Malik et al., 2021; Lindner et al., 2024; Liu et al., 2025). Despite these advances, neural models still struggle with generalization, particularly in social reasoning, as they often fail to capture the causal structure of behavior (de Haan et al., 2019; Codevilla et al., 2019; Bain & Sammut, 1995). This brittleness persists even with advanced techniques that learn contextual representations (Rabinowitz et al., 2018; Chuang et al., 2020; Jha et al., 2024) and does not disappear at scale under an assumption of imperfect rationality (Poddar et al., 2024). In contrast, our approach, which uses an LLM to generate open-ended code describing observed behavior, makes fewer assumptions about the nature of the agents being modeled. This allows it to capture everyday decision-making processes that may not be reward-maximizing.

**Large Language Models (LLMs) for Behavior Modeling.** LLMs may be a more effective bridge between the neural and symbolic paradigms. They enable enumerative inference for social reasoning (Wilf et al., 2023; Jung et al., 2024; Huang et al., 2024; Jin et al., 2024; Cross et al., 2024; Kim et al., 2025; Zhang et al., 2025), while neuro-symbolic frameworks (e.g., BIP + LLMs) improve robustness in embodied cooperation (Ying et al., 2024; Ding et al., 2024; Ying et al., 2025; Wan et al., 2025; Castro et al., 2025; Zhou et al., 2024; Qiu et al., 2024; Yang et al., 2024; Cao et al., 2024). However, existing implementations remain computationally intensive, often generating thousands of tokens for each prediction. In realistic settings, we need methods capable of rapid inference that still capture the structure of culturally shaped conventions and behaviors performed without deep cognitive processing (Bargh, 1994; Wood, 2024). By learning a code-based agent representation, ROTE avoids the high computational cost that BIP must incur to enumerate every possible goal.

**Program Induction.** Program synthesis has proven effective for world modeling (Guan et al., 2023; Wong et al., 2023b;a; Zhu & Simmons, 2024; Wong et al., 2025), action selection (Verma et al., 2021; Wang et al., 2023; Yao et al., 2023), and has even achieved near-expert performance on mathematical reasoning tasks such as International Math Olympiad problems (Trinh et al., 2024) while assisting researchers with science (Binz et al., 2025; Rmus et al., 2025). Neurosymbolic approaches, which combine LLMs or domain-specific neural networks with probabilistic program inference, have enabled agents to learn environment dynamics (Das et al., 2023) and master complex games like Sokoban and Frostbite with impressive sample efficiency (Tang et al., 2024; Tsividis et al., 2021; Tomov et al., 2023). Code-like representations have been used to infer reward functions from state-action transitions (Yu et al., 2023; Davidson et al., 2025), and LLMs have been harnessed to synthesize policies or planning strategies in domain-specific contexts (Liang et al., 2023; Sun et al., 2023; Trivedi et al., 2022). However, these prior approaches typically rely on well-defined rewards, domain-specific constraints, or focus on partial aspects of agent behavior, such as reward inference or demonstration summarization. In contrast, ROTE aims to infer an agent's causal decision-making process directly from observed behavior and assumes no access to reward signals or domain-specific structure.

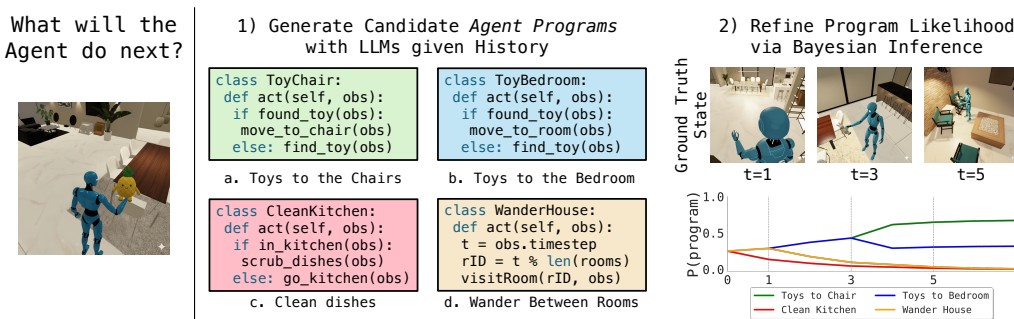

Figure 2: Overview of ROTE. ROTE predicts an agent's next action by generating and weighting Python programs that explain its observed behavior. From $t = 0$ to $t = 7$, ROTE observes a blue robot's trajectory. Initially, at $t = 1$, programs related to moving to the dining room are up-weighted. However, at $t = 3$, the robot picks up a toy, and ROTE remains uncertain if the goal is to clean up toys in the bedroom or place them on chairs in the living room. After the robot places the toy on a chair at $t = 5$, ROTE confidently updates its program weights to reflect the "bringing toys to chairs" script. By $t = 7$, ROTE can use this inferred script to rapidly and accurately predict future actions.

## 3 Representing Others' Trajectories as Executables

Drawing upon recent conceptualizations of "agents" in reinforcement learning and theoretical computer science (Abel et al., 2023a; Dong et al., 2021; Lu et al., 2023; Leike, 2016; Lattimore et al., 2013; Majeed & Hutter, 2018; Majeed, 2021; Cohen et al., 2019), we represent computationally bounded agents as programs with internal states, which can be conceptualized as Finite State Machines. This is formally represented using the notation $\lambda = (\mathcal{S}, s_0, \pi, u)$ from Abel et al. (2023b), where finite internal states $s_t \in \mathcal{S}$ used for decision-making in the policy $\pi : \mathcal{S} \to \Delta\mathcal{A}$ evolve via a transition function $u(s_{t-1}, a_{t-1}, o_t) \to s_t$, which maps the observations from the external world to the agent's next internal decision-making state. In the following section, we will demonstrate how we can search for the minimal program in the space of agents $\lambda \in \Lambda$ that best explains observed history of (observation $o \in \mathcal{O}$, action $a \in \mathcal{A}$) pairs, $h \in \mathcal{H}$. For the rest of this section, we use the notation $h_{0:t}$ to indicate the history of pairs from time 0 to $t$.

### 3.1 Agent Program Synthesis with Large Language Models

Given a finite length history $h_{0:t-1} \in \mathcal{H}$, from time 0 to $t - 1$, our objective is to find an agent $\hat{\lambda} \in \Lambda$ that both (1) takes the same action $a_t$ as the ground truth agent $\lambda^*$ when presented with observation $o_t$, and (2) minimizes its program size $|\hat{\lambda}|$. Encouraging concise program synthesis is not just a matter of engineering preference but is theoretically grounded in the foundations of algorithmic probability and inductive inference. Solomonoff's theory of inductive inference formalizes Occam's razor, demonstrating that the best scientific model for a given set of observations is the shortest algorithm (in terms of description length) that generates the data in question (Solomonoff, 1964; 1978; 1996). Under this framework, shorter programs are assigned higher prior probability, providing a universal solution to the problem of induction with strong convergence guarantees: the expected cumulative prediction error is bounded by the Kolmogorov complexity of the true data-generating process (Solomonoff, 1978; 1996). Thus, searching for minimal agent representations is not only computationally desirable but also theoretically optimal for generalization, a bias also observed in human programmatic reasoning (Bigelow & Ullman, 2025).

We operationalize our search through the space of agents $\Lambda$ with a two-stage approach: First, we optionally prompt an LLM to transform raw perceptual inputs into a natural language description of an agent's path. These percepts can be low-level observations like object coordinates in gridworlds, or even natural language scene-graphs from datasets like *Partnr* (Chang et al., 2025). Next, we have the LLM generate many possible Python programs

to obtain a distribution over possible code-based agent models which explain the observed behavior, $\Delta(\Lambda)$. Python is chosen for its readability, widespread use in AI research, and its power as a Turing-complete language, enabling the representation of arbitrarily complex decision-making logic in the worst case where $|\mathcal{S}| = |\mathcal{O}|$ for the ground-truth agent $\lambda^*$. Our prompting strategy makes two key assumptions: (1) the observed agent follows deterministic transitions between finite internal states $\mathcal{S}$ contingent on environmental/historical cues rather than executing complex adaptive policies, and (2) generated code should produce deterministic actions $a \in A$. Importantly, we ask the LLM to assume these properties of the observed trajectories *even if the ground truth agent generating the behavior is probabilistic and following sophisticated, goal-directed plans.* While this assumes a deterministic agent, we account for potential stochasticity in behavior with a noise model, allowing our approach to best approximate the underlying deterministic policy. We instruct the LLM to generate code that is efficient (low runtime complexity) and concise (minimize $|\lambda|$).

## 3.2 REFINING GENERATIONS THROUGH BAYESIAN INFERENCE

To form a more robust estimate of the true underlying agent program $\lambda^*$, we refine the distribution over candidate programs $\Delta(\Lambda)$ obtained from the language model using Sequential Monte Carlo. Specifically, we estimate the posterior probability of a candidate agent program $\lambda$ given the observed history $h_{0:t-1}$ using the relationship:

$$p(\lambda|h_{0:t-1}) \propto p(h_{0:t-1}|\lambda)p(\lambda). \tag{1}$$

This approach is related to inverse planning-based methods that infer latent goals given observed behavior (Ullman et al., 2009; Baker et al., 2017; Shum et al., 2019; Netanyahu et al., 2021). However, instead of assuming a fixed, often complex, planner (like MCTS or brute-force search) and performing inference over a space of goals, our method condenses all behavioral conventions and scripts an agent might follow into a single programmatic representation $\lambda$. Since $\lambda$ is a deterministic program, we give the action $\hat{a}_t$ it predicts the ground-truth agent will take at observation $o_t$ a probability of $(1 - \epsilon)$ and all other actions $a^- \in \mathcal{A} - \{\hat{a}_t\}$ a probability of $\frac{\epsilon}{|\mathcal{A}|-1}$. This effectively allows $\lambda$ to predict a distribution over actions $\Delta(\mathcal{A})$ it might take at each step. Then, we can perform inference directly over the space of likely decision-making processes encoded as Python programs by calculating $p(\lambda|h_{0:t-1}) \propto \Pi_{o_i,a_i \in h_{0:t-1}} p(a_i|o_i, \lambda) \cdot p_{\text{prior}}(\lambda)$. With this refined posterior distribution, we select the $k$ most likely agent programs, and execute the corresponding Python code for each from the current observation $o_t$. Then, ROTE performs a weighted combination of agent programs to form our approximation $\lambda^* \approx \hat{\lambda} = \sum_{\lambda \in \Delta(\Lambda)} p(\lambda|h_{0:t-1}) \cdot \lambda(\cdot|o_t)$.

The combination of LLM-based program synthesis with Bayesian Inference results in our method for inferring others' behaviors, **ROTE**. Pseudocode for our approach can be found in Algorithm 1, and in Figure 2, we provide an overview of ROTE on an intuitive example in the *Partnr* environment, an embodied robotics simulator where an agent tries to help a human complete a variety of household chores (Chang et al., 2025). We additionally include examples of agent code inferred by ROTE in *Construction* and *Partnr* in Appendix A.11.2.

## 4 EXPERIMENTS

**Environments.** We evaluate ROTE across two distinct environments. First, we use *Construction* (shown in Figure 7), a fully-observable 2D grid-world where agents actively navigate obstacles like walls and other agents, and can transport colored blocks to different locations on the map (Jha et al., 2024). Then, we explore the efficacy of our method on *Partnr* (shown in Figure 5), a large-scale embodied robotics simulator where an AI-assistant perceives a realistic home or office space as a natural language scene-graph (Chang et al., 2025). Built on the *Habitat* benchmark, this environment requires the agent to utilize tools to help a human complete tasks in a partially observable world (Puig et al., 2023).

**Baselines.** We compare ROTE against three baselines: **Behavior Cloning (BC)**. In the *Construction* environment, the BC model is a neural network with an LSTM trained on pixel-based observations of agent trajectories (Rabinowitz et al., 2018); for *Partnr*, we fine-tuned Llama-3.1-8b to imitate a ground-truth LLM agent's behaviors using a training

set of (scene-graph, action) pairs (Chang et al., 2025). **Automated Theory of Mind (AutoToM)** (Zhang et al., 2025). AutoToM is a neuro-symbolic approach which uses LLMs to generate open-ended hypotheses about an agent's beliefs, goals, and desires, then applies Bayesian Inverse Planning to find the most likely action. **Naive LLM (NLLM)**. NLLM simply prompts an LLM with observed states and environment dynamics to predict the next action directly. Our evaluation for all methods except for BC uses a suite of LLMs: Llama-3.1-8b Instruct, DeepSeek-V2-Lite (16b), DeepSeek-Coder-V2-Lite-Instruct (16b), and we report the highest accuracy achieved for each baseline to ensure the most competitive comparison. All results for ROTE were obtained using DeepSeek-Coder-V2-Lite-Instruct, while other baselines show the highest-performing model for each environment. Appendix A.7 provides a detailed breakdown of per-task and per-LLM accuracy for all methods, demonstrating our approach's consistent success across different LLM model types.

**Dataset Generation.** For the fully observable *Construction* environment, we hand-designed 10 distinct Finite State Machines to generate $50,000$ state-action pairs across 100 trajectories/agent $\times 10$ agents = 1000 trajectories. Behaviors ranged from simple tasks, such as patrolling, where agents rely on planning heuristics, to complex goal-directed tasks using A* search, like finding all green blocks. We have included some for illustration in Figure 7 and the full list of behaviors in Appendix A.1. For the partially observable *Partnr* environment, we used the LLM agents defined in (Chang et al., 2025) to generate state-action pairs for a robot assistant completing diverse tasks (i.e. "clean all toys in the bedroom") from their "train" and "validation" datasets. In these datasets, states are represented as natural language scene graphs, and actions are high-level tools.

**Evaluation Protocol.** We evaluate using two protocols: (1) single-step prediction, where given observations from timesteps 0 to $t$, the task is to predict the action $a_t$; and (2) multi-step prediction, where we iteratively predict actions $\hat{a}_t, \ldots, \hat{a}_{t+10}$ conditioned on the ground-truth observed states $o_0, \ldots, o_t$. For the BC model in *Construction*, we hold out 100 trajectories for evaluation, training on the remaining data. All baselines are evaluated on these 100 held-out trajectories. For *Partnr*, we evaluate single-step prediction with $t = |\mathcal{H}| - 2$, since varying trajectory lengths make multi-step evaluation inconsistent, and the final timestep is always the terminal action. We evaluate all models on the entire "validation" dataset, using the "train" dataset to finetune the BC model. We only predict high-level tools used by agents in *Partnr*, since AutoToM requires static-sized action spaces (Zhang et al., 2025).

**Human Studies.** We conducted human studies in the single-agent *Construction* environment to evaluate ROTE's ability to predict human behavior and to benchmark its performance against human predictions. For the first study, 10 participants were recruited to perform their interpretation of each of the 10 handcrafted FSMs without observing the ground-truth code, generating 30 state-action pairs/person/script. In a separate study, we recruited 25

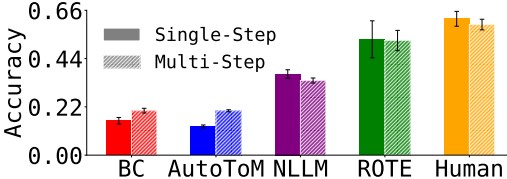
(a) Single-step vs. multi-step prediction accuracy for *Construction* with **scripted** agents

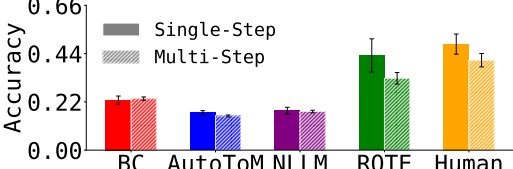
(b) Single-step vs. multi-step prediction accuracy for *Construction* with **human** agents

Figure 3: ROTE outperforms all baselines in both single-step and multi-step action prediction for scripted (a) and human agents (b). ROTE's code-based representations, which treat human actions as efficient scripts, enable it to generalize effectively from limited observations. For single-step predictions, ROTE was significantly more accurate than all baselines for both scripted ($p < 0.05$ for NLLM, $p < 0.001$ for BC and AutoToM) and human agents ($p < 0.05$ for BC, $p < 0.01$ for NLLM, $p < 0.001$ for AutoToM). This superior performance was maintained in multi-step predictions for both agent types (scripted: $p < 0.001$ for BC, AutoToM, and NLLM; human: $p < 0.01$ for BC, $p < 0.001$ for NLLM and AutoToM). *ROTE achieved human-level predictive accuracy of human behavior.*

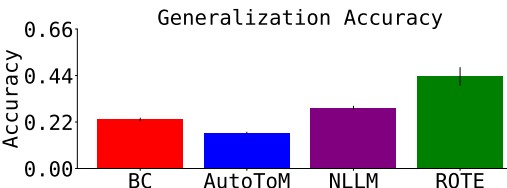

Figure 4: ROTE demonstrates superior zero-shot generalization to novel environments in *Construction*. Without any additional conditioning on an agent's behavior, the programs ROTE infers from one environment transfer to novel settings more effectively than all other baselines ($p < 0.001$ in a two-sided t-test).

humans to act as predictors. They were shown a human's trajectory from $t = 0$ to $t = 19$ and the state at $t = 20$, and asked to predict its next five actions, from $t = 20$ to $t = 24$. We use the same setup for a third study to explore how well people predict the behavior of the ground-truth FSM's next actions instead. We compared peoples' prediction accuracy to ROTE and the other baselines to benchmark different behavior modeling algorithms. All studies were approved by our university's Institutional Review Board (IRB) and were designed using NiceWebRL (Carvalho et al., 2025). We used Prolific for crowdsourcing data. We plan to open-source the code for our baselines, datasets, and human evaluations.

## 5 RESULTS

**How well does ROTE model and predict scripted agent behavior?** To evaluate the effectiveness of ROTE, we first examined its predictive accuracy in a controlled setting where agents in the *Construction* environment followed one of 10 handcrafted programs. These programs were not provided to ROTE at any point during evaluation. Our results in Figure 3a demonstrate that ROTE consistently surpasses all baselines in both single-step and multi-step prediction accuracy in this evaluation setting and *does not statistically significantly underperform human performance* (p=0.3087 for single-step and p=0.1679 for multi-step in a two-sided t-test). While these initial results were promising, a potential concern was that ROTE might simply be exploiting repetitive patterns, rather than learning the underlying policy. We investigated this by measuring how often an agent revisited a state or repeated an action. We found an *extremely low* correlation between ROTE's accuracy and either of these metrics (0.303 for matching states, 0.064 for matching actions), confirms that ROTE is not exploiting simple data regularities. This finding, paired with ROTE's strong multi-step performance, suggests that code-based representations can be effective for learning the underlying policies that enable robust, long-term predictions.

**How well does ROTE model and predict human behavior?** Having established that ROTE's code-based representations are effective in controlled, scripted environments, we next wanted to test its ability to model more complex, nuanced behaviors. We began by evaluating ROTE against human agents performing 10 tasks in the *Construction* environment. As illustrated in Figure 3b, ROTE outperforms all baseline algorithms and **achieves human-**

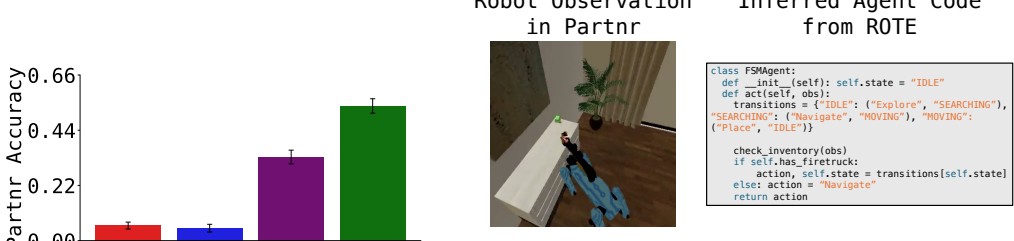

(a) Action prediction accuracy in *Partnr*    (b) Example *Partnr* task with ROTE's inferred program

Figure 5: (a) Prediction accuracy in the large-scale, partially observable *Partnr* environment. ROTE demonstrated a superior ability to anticipate the behavior of goal-directed, LLM-based agents, with a two-sided t-test showing ROTE significantly outperformed all other models ($p < 0.001$). (b) The pseudocode example illustrates how ROTE's inferred programs capture complex task logic using conditionals and state-tracking.

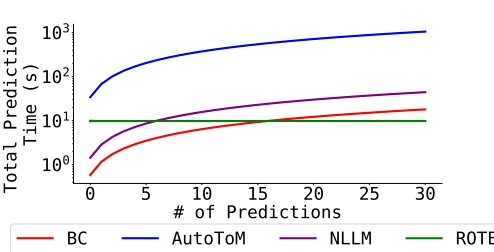

Figure 6: Total multi-step prediction time in *Construction*. Despite being slower than BC and Naive LLM prompting in the single-step prediction case, ROTE's programmatic representations enable its multi-step compute cost to scale orders of magnitude more efficiently than other approaches, making it better suited for long-horizon settings than other approaches for predicting individual behavior.

**level predictive accuracy of next-step human actions**. A deeper per-task accuracy analysis, shown in Figure 9, reveals that ROTE has greater accuracy than humans on some tasks with repetitive patterns, such as "move up if possible, otherwise down" or "move in an L-shape." However, humans are still much better at anticipating scripts for tasks such as "patrol the grid clockwise" and goal-directed tasks such as "move all pink blocks to the corner of the grid." This gap highlights that while the code produced by ROTE is expressive enough to capture many behaviors, more powerful LLMs with enhanced reasoning may be needed to achieve human-level prediction in all settings.

**Generalizing to Novel Environments:** A key advantage of modeling behavior with scripts is the potential for rapid generalization to new, but similar, environments. We wanted to know if ROTE's inferred programs could transfer without needing to be relearned. To test this, we first observed a scripted agent following a pattern like "patrol counterclockwise" for 20 timesteps, and then showed the same agent in a distinct environment. We then asked ROTE and the baselines to predict the next 10 actions of the same agent. For ROTE, this was done by using the same set of programs inferred in the first environment for prediction without updating their likelihoods. Figure 4 shows that ROTE can still predict the agent's behavior accurately in the new setting, outperforming all baselines without needing to re-incur the cost of text generation, a necessary step for NLLM and AutoToM. Although ROTE's performance decreases compared to predicting behavior in the original environment in Figure 3a, its ability to generalize makes it a more accurate and efficient alternative to traditional Inverse Planning or purely neural methods.

**Can ROTE's code-based approach scale to model behavior in complex, realistic environments?** To further push the boundaries of ROTE's capabilities, we tested it on the embodied robotics benchmark *Partnr*, where the task is to predict the next tool utilized by LLM-agents simulating a human or robot completing chores. This environment is particularly challenging due to partial observability and long-horizon, compositional tasks such as "find a plate and clean it in the kitchen" or "look for toys and organize them neatly in the bedroom." Despite this complexity, Figure 5 shows our approach significantly outperformed all baselines, including inverse planning and behavior cloning methods, and those incorporating LLMs. To better understand the types of problems ROTE excels at, we used Llama-3.1-8B-Instruct to cluster the ground-truth tasks from our test set into three categories, as shown in Figure 10. While baselines like AutoToM and Behavior Cloning showed success with tasks involving simple navigation, ROTE demonstrated a superior ability to handle more intricate problems, such as turning items on/off and cleaning objects. This demonstrates its generalizability in creating code for agents that face uncertainty and possess beliefs about their environment.

**How does the computational efficiency of ROTE compare to other approaches?** Forming long-horizon plans in the presence of other agents requires predicting their behaviors over time quickly and not just accurately. To understand whether ROTE scales effectively, we plot the time in seconds required for different baselines to make predictions about agents' behaviors multiple times into the future in *Construction*. As shown in Figure 6, while ROTE is initially slower for single-step predictions compared to BC and NLLM baselines due to the need to generate and prune candidate programs, its *test-time compute costs scale orders of magnitude more efficiently with the number of predictions*. This is because once ROTE's code-based representations are inferred, it can execute these programs rapidly for all future steps. In contrast, other LLM-based methods must re-generate a response for every new

time step. We analyze additional factors contributing to this efficiency in Appendix A.5 and Figure 14. Taken together with the results from Figures 3, 4, and 5, this illustrates that code-based representations can balance predictive power with prediction efficiency.

**What is the relationship between ROTE's core components and its predictive performance?** To understand how ROTE achieves its superior generalization and human-level accuracy, we conducted a series of ablation studies on its core components. We found that ROTE's two-stage observation parsing, which converts observations into a natural language description before generating code, had a minimal effect on accuracy for the FSM and human gameplay datasets in *Construction* (Figure 11). However, this process significantly hurt performance in *Partnr*. This is likely because *Partnr*'s observations are already rich scene graphs (Chang et al., 2025), and the abstraction step removes crucial details needed for effective program generation. Additionally, we investigated the use of Sequential Monte Carlo (SMC) with rejuvenation versus standard Importance Sampling. SMC, which replaces low-likelihood programs with new ones, improved early-stage accuracy when the number of sampled hypotheses was small (Figure 12). This benefit, however, diminished as the initial set of candidate hypotheses increased, suggesting that the initial diversity provided by the LLM is often sufficient.

Lastly, we analyzed the impact of imposing different degrees of structural constraints on ROTE's program generation, inspired by methods for inferring reward functions (Yu et al., 2023). We evaluated three variants: "Light" (assuming agents are FSMs without providing examples), "Moderate" (defining FSM states explicitly but allowing open-ended code), and "Severe" (a two-stage process converting natural language predictions of FSMs into code). Our previous results were based on the "Light" condition. The optimal level of structure, however, varied by environment, as shown in Figure 13. In the *Construction* environment, where agents followed predictable FSMs, the Severe approach performed as well as others. This suggests that for predictable, rote behaviors, an explicitly structured representation can be just as effective while also being computationally efficient (Callaway et al., 2018; Lieder & Griffiths, 2020; Callaway et al., 2022; Icard, 2023). Conversely, modeling human behavior proved less suited for strict FSMs. The Moderate condition was superior for human gameplay, highlighting the need for representational flexibility when agents are following a general script but exhibiting inherent variability. In the partially observable *Partnr* environment, forcing agents into a strict FSM representation performed significantly worse than open-ended code generation, suggesting these scenarios might be better suited for traditional Inverse Planning methods that can handle a wider range of states and tasks. These findings reveal a gradient of agentic representations, from automatic to goal-directed, which allows for flexible prediction across different scenarios. Future work could use meta-reinforcement learning to dynamically select the appropriate level of representational structure based on the task.

## 6 DISCUSSION

In this work, we framed behavior inference as a program synthesis problem, showing that our approach, ROTE, can accurately and efficiently predict the actions of machines *and real people* in complex environments. ROTE offers a scalable alternative to traditional methods that require extensive datasets or significant computational resources. This has immediate implications for domains where real-time adaptability and interpretability are crucial, such as with caregiver robots that could use ROTE's representations to anticipate daily routines.

**Limitations:** While our results highlight the effectiveness of program synthesis for text-based observations, we note the limitations of the applicability of our findings in *Partnr* since we only predicted high-level tools used by agents, which was done to accommodate baselines which required static action spaces. While our evaluation in *Partnr* still involved more tools than our other experiments (19 actions in *Partnr* compared to 6 in the *Construction*), future research should explore ROTE in high-dimensional, continuous control settings. In those cases, ROTE might need to be integrated with vision-language models (VLMs) to parse pixel-based inputs (e.g., raw video feeds for assistive robots) and neural control mechanisms to execute plans, effectively operating at the level of option prediction (Sutton et al., 1999). Another interesting direction would be to explore how the size of LLMs used

for behavioral program inference impacts prediction quality in more sophisticated scenarios, such as modeling team coordination in workplaces or norm enforcement on social platforms.

Lastly, unlike traditional Theory of Mind approaches that predict beliefs and goals, our work focuses solely on action prediction. If we view beliefs as dispositions to act (Ramsey & Moore, 1927; Ryle, 1949), predicting a distribution over an agent's internal decision-making states and logic for transitioning between them is functionally equivalent to belief inference. ROTE is designed to excel in scenarios dominated by predictable, routine, or script-like behaviors, such as daily routines in warehouses and stores, relatively stable social conventions like driving, or routine household settings. This is because ROTE exploits the efficiency of executing simple code for long-horizon prediction in these routine settings. For ROTE to gain true generality and address the rigidity concern, future work is explicitly focused on extending it to generate Probabilistic Programming Languages (PPLs), such as *memo*, which is specialized for social reasoning in JAX (Chandra et al., 2025). This extension would allow ROTE to infer the distribution over actions or latent mental states, directly addressing the stochastic nature of human actions without abandoning the executable code format. In terms of failure modes, domains requiring high-fidelity continuous control over raw sensor data (e.g., video feeds) require ROTE's inferred high-level programs to be integrated into a Task and Motion Planning architecture, where ROTE provides the symbolic task plan to a low-level neural control mechanism. Finally, for deeply complex, goal-directed behaviors involving "unknown unknowns" in partially observable environments, the very notion of a fixed FSM-like programmatic model may be fundamentally unsuitable, indicating that in these cases, the representation itself is too rigid to capture the agent's full intentionality. Thus, we view ROTE as generating and reasoning over one of many possible representations that are suitable for behavior prediction rather than a catch-all.

## ACKNOWLEDGEMENTS

We would like to thank the Cooperative AI Foundation, the Foresight Institute, the UW-Amazon Science Gift Hub, the Sony Research Award Program, UW-Tsukuba Amazon NVIDIA Cross Pacific AI Initiative (XPAI), the Microsoft Accelerate Foundation Models Research Program, Character.AI, DoorDash, and the Schmidt AI2050 Fellows program for their generous support of our research. This material is based upon work supported by the Defense Advanced Research Projects Agency and the Air Force Research Laboratory, contract number(s): FA8650-23-C-7316. Any opinions, findings and conclusions, or recommendations expressed in this material are those of the author(s) and do not necessarily reflect the views of AFRL or DARPA. Lastly, we would like to express our gratitude to our colleagues at the Social Reinforcement Lab and the Computational Minds and Machines Lab, as well as Guy Davidson, Tan Zhi-Xuan, and Tomer Ullman for inspiring conversations and insights during the course of this project.

## IMPACT STATEMENT

This paper presents a framework for modeling agentic behavior through program synthesis, offering a scalable and interpretable alternative to traditional neural and symbolic methods for action prediction. The primary societal benefit of this work lies in improving the safety and efficiency of human-AI collaboration in real-world settings, such as assistive robotics, where anticipating human routines is essential for effective support. By representing behavior as executable Python code, ROTE provides a layer of transparency that allows users to inspect and verify the AI's underlying assumptions about human logic, which is essential for building trust in autonomous systems. However, we acknowledge that the ability to infer high-fidelity behavioral "scripts" from sparse data carries potential risks regarding privacy and the unauthorized profiling of individuals. Furthermore, developers must be cautious of "script rigidity," where a system might fail to account for stochastic human variability in safety-critical edge cases. We believe that by transitioning toward interpretable and eventually probabilistic programmatic representations, our work advances the field of Machine Learning toward social intelligence that is both technically robust and ethically accountable.

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

# A  Appendix

## A.1  Ground Truth Agent Behaviors for *Construction*

For research question 1 in Section 5, we hand designed 10 agents, represented as Finite State Machines, to engage in diverse behaviors. The agents varied in complexity, with some using sophisticated A-star search to achieve a goal, and others using faster, less resource-intensive planning heuristics, namely the Manhattan distance as an approximation of how valuable an action is for an agent looking to move to a target location. While there is a large body of literature and debate surrounding what it means to be goal-directed, in this work we say any agents conducting forward plans, denoted by explicit rollouts within the environment, are considered to be goal-directed. In the *Construction* gridworld task, this means the agents using A-star to complete tasks such as "pickup green blocks and move them to the corner" and "pair all blue blocks together" are goal-directed, whereas "patrol the grid in a clockwise direction," which uses the Manhattan distance and FSM states as planning heuristics, are considered scripted. In *Partnr*, all of the LLM agents are goal-directed since they use the ReAct framework to plan how to complete household tasks. When collecting human data, we do not know if the participants are conducting detailed planning. We note that while we motivate our work from prior literature in cognitive science about predicting the behavior of scripted agents, our empirical results demonstrate that our approach is robust to predicting goal-directed behavior in the sense that we define it here. We summarize the behaviors and internal decision making states for all ground-truth agents below:

1. **Block Cycle**: Using the manhattan distance as the planning heuristic, move from the green block to the blue block to the purple block to the green block and so on. If the agent ever has a block in its inventory, it will immediately drop it and resume its cycling behavior.

2. **Clockwise Patrol**: If an agent is not along the outermost wall of the grid, it will repeatedly alternate between moving left and moving up until it hits a wall. Then, it will follow the wall clockwise: if there is a wall above it, the agent will move right repeatedly until it hits a wall, then repeats this process for going down, left, up and right again. If the agent ever has a block in its inventory, it will immediately drop it and resume its cycling behavior.

3. **Counter-clockwise Patrol**: This agent is the same as Clockwise Patrol, except it it will patrol the border wall in a counter-clockwise manner, moving left repeatedly until it hits a wall, then doing the same for moving down, right, up and left again.

4. **Left-Right Patrol**: The agent will move left until it hits a wall, then will move right until it hits a wall, and repeat this process. If the agent ever has a block in its inventory, it will immediately drop it and resume its patrolling behavior.

5. **Pair Blue Blocks**: This agent uses A-star search for planning. If it does not have a blue block in its inventory, it finds and executes the shortest path to a blue block. Then, it uses the "interact" action to add the block to its inventory, and uses A-star to find the shortest path to a different blue block.

6. **Patrol with A-Star**: Here, the agent's goal is to repeatedly cycle between the top left, top right, bottom right, and bottom left corners of the grid. While Clockwise Patrol has a behavior which on the surface may seem similar, for Patrol with A-star we introduced addition complexity by having the agent believe it incurs a penalty for touching any of the colored blocks. As such, it uses A-star with negative edge values given to any action which leads an agent to landing on a colored block, thus resulting in behavior which tries to patrol but frequently leaves and returns to the border to

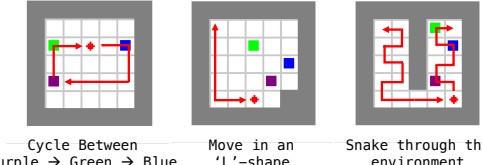

Figure 7: Example scripts from *Construction.* We designed a suite of goal-directed (planner-based) and automatic (heuristic-based) agentic behaviors, from patrolling to transporting specific blocks to a location.

avoid colored blocks. Again, if it ever picks up a colored block, it immediately drops it.

7. **L-shaped Patrol**: This agent, initially at a coordinate $(x, y)$, will move down until it collides with a wall, then will move right until it collides with a wall. Then, it will return to its original location, first moving left until its x-coordinate is $x$, and up until its final coordinate is $(x, y)$. It repeatedly does this process. The agent immediately drops any blocks in its inventory.

8. **Transport Green**: Here, the agent uses A-star search to move towards a green block and pick it up. Then, it uses A-star search to move the green block as close to an empty corner grid cell.

9. **Snake Patrol**: This agent has four internal decision-making states: 1) Moving down/right, where the agent moves right until it cannot any more, then moves down one step; 2) Moving down/left, where the agent moves left until it cannot any more, then moves down one step; 3) Moving up/right, where the agent moves right until it cannot any more, then moves up one step; 4) Moving up/left, where the agent moves left until it cannot any more, then moves up one step. The resulting pattern appears like a snake moving throughout the grid.

10. **Up/Down Patrol**: The agent will move up until it hits a wall, then will move down until it hits a wall, and repeat this process. If the agent ever has a block in its inventory, it will immediately drop it and resume its patrolling behavior.

## A.2    HUMAN RESULTS BREAKDOWN

In Figure 8, we show the accuracy for ROTE compared to humans when predicting FSM behavior in *Construction.* An example of some of the task are shown in Figure 7. In Figure 9, we show the accuracy for ROTE compared to humans when predicting Human behavior in *Construction.* We find that humans excel at predicting goal-directed tasks while our method performs better with repetitive tasks, although all of the variance in predictive accuracy cannot be captured by this distinction. In subsequent followups, we plan to do a greater exploration of the different error modes of humans and other models, as well as scale ROTE to larger language models, to see whether ROTE is an accurate computational model of human behavior.

## A.3    CLUSTERED TASK BREAKDOWN IN *PARTNR*

To understand the types of tasks ROTE excels at compared to baselines in the *Partnr* simulator, we used Llama-3.1-8B-Instruct to cluster the ground-truth tasks from our test set into three categories. As shown in Figure 10, we report the mean prediction accuracy and standard error for each algorithm on a per-cluster basis. While AutoToM and Behavior Cloning show some success on tasks involving simple actions like moving and rearranging objects, they struggle significantly with more complex interactions, such as turning items on/off or cleaning. ROTE, in contrast, maintains a degree of accuracy in these more challenging settings.

## A.4    MODEL COMPONENT ANALYSIS

We show the effect of different model components. While the choice of observation parsing did not have too much of an impact on the *Construction* evaluations, Figure 11 indicates it has a significant effect on predictive performance in *Partnr*. This is likely because the

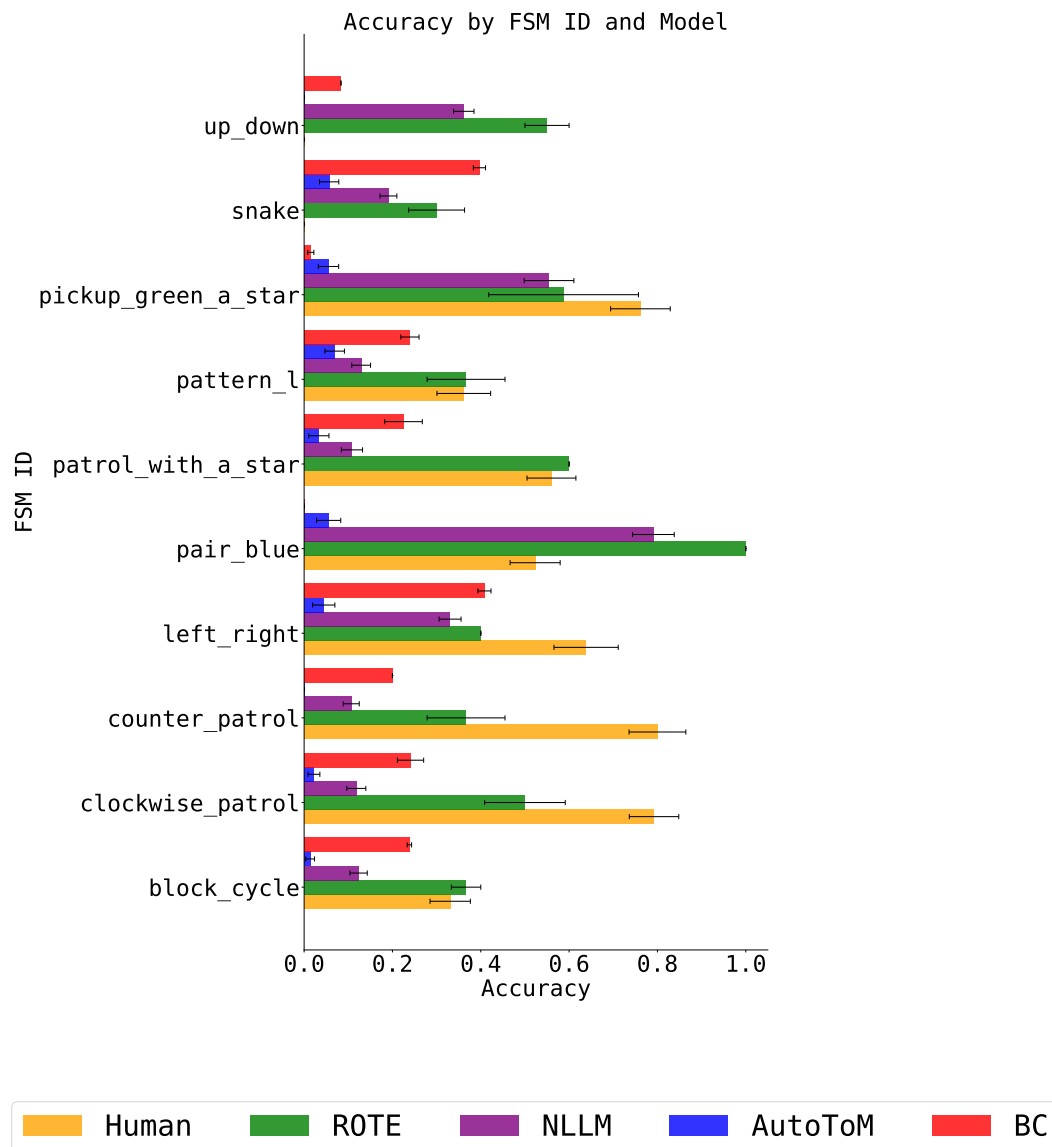

Figure 8: Per-task accuracy comparison between different methods predicting ground truth FSM gameplay.

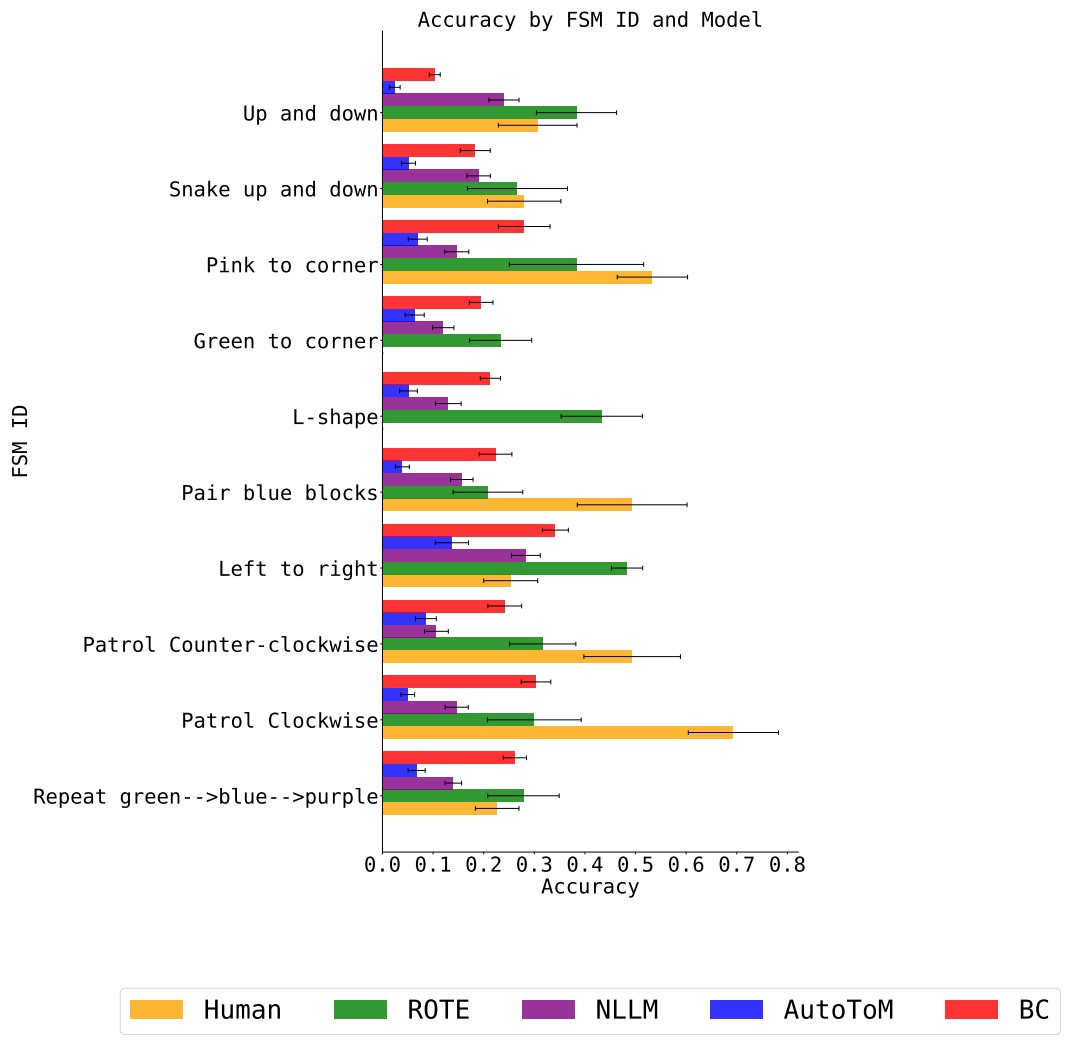

Figure 9: Per-task accuracy comparison between different methods predicting human gameplay. While ROTE succeeds at more routine tasks, humans excel in predicting more goal directed behaviors.

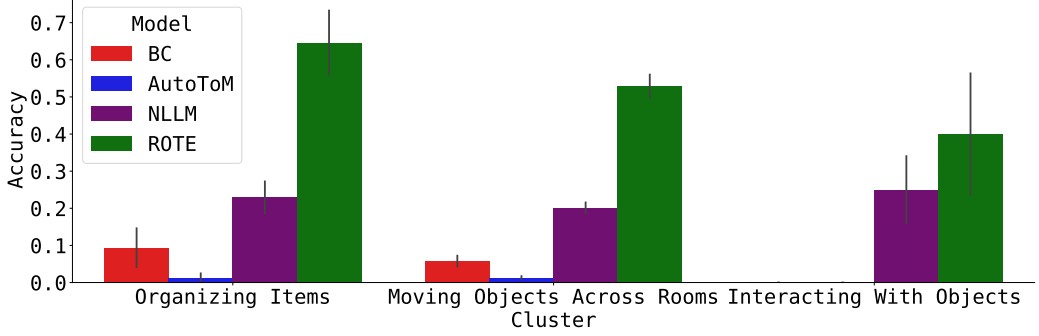

Figure 10: Task-specific generalization in *Partnr*. We used Llama-3.1-8B-Instruct to cluster our prediction tasks into three distinct categories. We report the mean accuracy and standard error (SE bars) for each algorithm. While baselines like AutoToM and Behavior Cloning perform adequately on tasks involving object manipulation, they struggle with more complex interactions. ROTE, however, maintains performance on these more intricate problems.

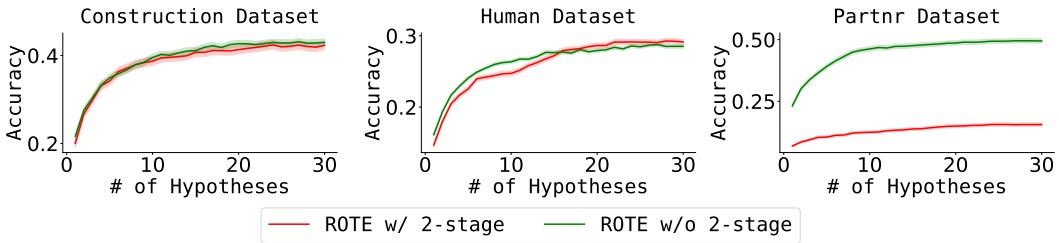

Figure 11: Ablating Observation Parsing

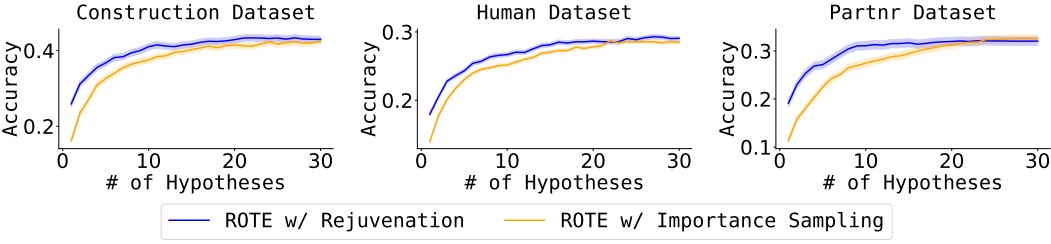

Figure 12: Ablating Inference Algorithm

observations, which are already in natural language, contain critical information on the data structures they are represented as that abstraction removes.

Figure 12 demonstrates the benefits of different inference algorithms in ROTE. While ROTE is not very sensitive to the choice of probabilistic inference method used as it has more candidate agent programs, if agents are constrained by the number of hypotheses they can maintain, performing SMC with rejuvenation proves to be a more effective strategy, since this effectively augments the number of programs considered.

Figure 13 reveals an interesting gradient along which different degrees of structure influence ROTE's ability to predict behaviors. In controlled settings where agents are Finite State Machines following deterministic transitions between behaviors, increasing the amount of structure used to predict what they will do next does not significantly harm performance. This can be a useful inductive bias that reduces cognitive load for agents interacting with systems that require prediction in order to effectively interact with, such as a thermostat, but are nevertheless simple enough to represent as a series of rules. In the human-behavior setting, this does not hold as well. We find a moderate amount of structure, where providing more detailed examples about what the internal mechanisms of the observed agent look like without forcing ROTE to generate code following that structure, performs the best. These settings are closest to realistic encounters with other people: when walking down the street or ordering coffee, we may try to follow scripts or conventions for how to interact, but there is inherent variability in our behaviors that more open-ended programs must account for. Lastly, when predicting the behavior of agents that are goal-directed in a partially observable world, imposing FSM structure greatly diminishes performance. These are scenarios where prediction might best be performed by more complex reasoning processes about an agent's intentions and beliefs. Here, constraining code to be structured as an FSM might fail to account for how agents react to the presence of unknown unknowns they encounter.

## A.5 Relationship between Program Size ($|\lambda|$) and Accuracy

As shown in Figure 14, higher prediction accuracy in *Construction* and *Partnr* corresponds to shorter programs (in characters). This occurs *even though program length is not explicitly factored into the likelihood computation*, suggesting that the approach naturally favors a simple, efficient representation of the agent's behavior. This aligns with our hypothesis, inspired by Solomonoff (Solomonoff, 1964; 1978; 1996), that shorter programs will generalize more effectively due to Occam's razor.

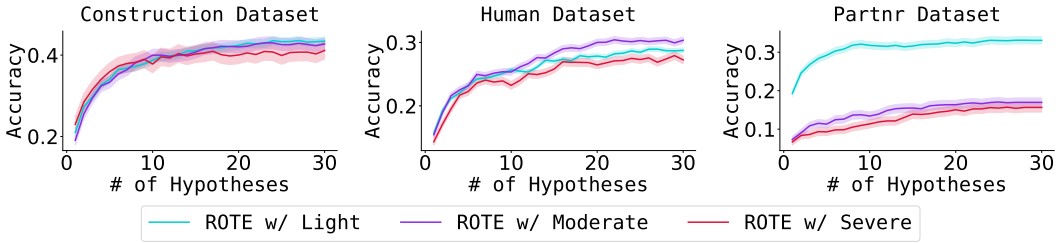

Figure 13: Ablating Structure Enforced in Generated Code

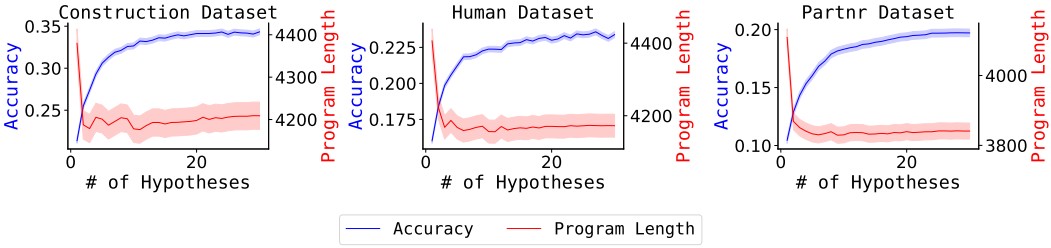

Figure 14: Average program length (in characters) versus prediction accuracy as a function of the number of generated hypotheses for *Construction* and *Partnr*. Shorter programs yield higher accuracy for scripted, human, and LLM agents.

## A.6 Top-k Effect

In Figure 15 we explore the impact of different $k$ values for the top-k hypothesis pruning phase after generation. We tried $k = 1$, 10, and 30. We did not find any meaningful variation in performance as a function of $k$. This suggest the choice of which hyperparameter to use may be left to the agent designer. Whereas smaller $k$ values enable faster inference, larger values enable better uncertainty estimation. Moreover, because of the largely deterministic nature of the generated programs, there can be an implicit top-$k$ effect at higher hypothesis numbers, wherein unlikely programs are assigned very low probabilities throughout a trajectory, effectively leading to their pruning during policy selection for action prediction.

## A.7 Per-llm Results

In Tables 1, 2 and 3, we report the raw accuracy of different LLM models using different algorithms, as well as the standard error, on the Scripted, Human, and LLM-agent behavior datasets in *Construction* and *Partnr*. For the results reported in the paper, we had to tune the number of hypothesis and other hyperparameters, such as whether to use two-stage observation parsing, on a dataset-by-dataset basis. We did this by running a sweep of

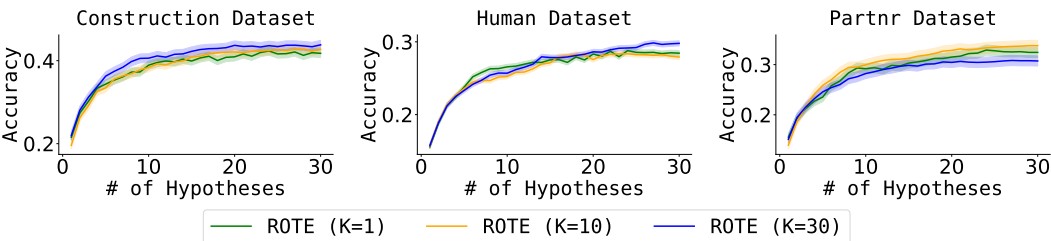

Figure 15: Top-$k$ parameter analysis in *Construction*. No appreciable difference in accuracy as a result of different parameters, suggesting the choice between uncertainty preservation (maintaining a larger set of hypotheses from a larger $k$) and prediction speed (by executing less programs with a smaller $k$) is up to the agent and agent designer.

hyperparameters and comparing their performance on 20% of the data, then utilizing the best performing hyperparameter from that subset, as the selected model configuration for the remaining 80% of the data. The hyperparameters used for each environment can be found in Section A.11.

| Algorithm | DeepSeek-Coder-V2-Lite-Instruct (16B) | DeepSeek-V2-Lite (16B) | Llama-3.1-8B-Instruct |
|---|---|---|---|
| AutoToM | $0.000 \pm 0.000$ | $0.000 \pm 0.000$ | $0.202 \pm 0.023$ |
| NLLM | $0.310 \pm 0.032$ | $0.266 \pm 0.018$ | $0.340 \pm 0.033$ |
| Chain-of-Thought | $0.210 \pm 0.04$ | $0.210 \pm 0.04$ | $0.269 \pm .025$ |
| ROTE (light) | $0.479 \pm 0.033$ | $\mathbf{0.312 \pm 0.032}$ | $\mathbf{0.477 \pm 0.044}$ |
| ROTE (moderate) | $0.436 \pm 0.042$ | $0.256 \pm 0.032$ | $0.446 \pm 0.051$ |
| ROTE (severe) | $0.457 \pm 0.037$ | $0.298 \pm 0.033$ | $0.390 \pm 0.049$ |
| ROTE (two-stage) | $\mathbf{0.522 \pm 0.046}$ | $0.271 \pm 0.028$ | $0.468 \pm 0.052$ |

Table 1: Multi-step LLM results (with standard error) for Ground-truth Scripted Gameplay Data Prediction in *Construction*.

| Algorithm | DeepSeek-Coder-V2-Lite-Instruct (16B) | DeepSeek-V2-Lite (16B) | Llama-3.1-8B-Instruct |
|---|---|---|---|
| AutoToM | $0.000 \pm 0.000$ | $0.000 \pm 0.000$ | $0.156 \pm 0.011$ |
| NLLM | $0.151 \pm 0.012$ | $0.176 \pm 0.013$ | $0.171 \pm 0.016$ |
| Chain-of-Thought | $0.000 \pm 0.000$ | $0.000 \pm 0.000$ | $0.156 \pm 0.011$ |
| ROTE (light) | $0.296 \pm 0.019$ | $0.199 \pm 0.015$ | $0.305 \pm 0.022$ |
| ROTE (moderate) | $0.310 \pm 0.018$ | $0.204 \pm 0.021$ | $0.266 \pm 0.024$ |
| ROTE (severe) | $0.304 \pm 0.022$ | $\mathbf{0.230 \pm 0.018}$ | $0.245 \pm 0.026$ |
| ROTE (two-stage) | $\mathbf{0.329 \pm 0.031}$ | $0.209 \pm 0.014$ | $\mathbf{0.327 \pm 0.026}$ |

Table 2: Multi-step LLM results (with standard error) for Human Gameplay Data Prediction in *Construction*.

| Algorithm | DeepSeek-Coder-V2-Lite-Instruct (16B) | DeepSeek-V2-Lite (16B) | Llama-3.1-8B-Instruct |
|---|---|---|---|
| AutoToM | $0.000 \pm 0.000$ | $0.000 \pm 0.000$ | $0.050 \pm 0.015$ |
| NLLM | $0.113 \pm 0.018$ | $0.333 \pm 0.027$ | $0.170 \pm 0.022$ |
| Chain-of-Thought | $0.000 \pm 0.000$ | $0.000 \pm 0.000$ | $0.050 \pm 0.015$ |
| ROTE (light) | $\mathbf{0.537 \pm 0.029}$ | – | $0.439 \pm 0.066$ |
| ROTE (moderate) | $0.472 \pm 0.029$ | $0.026 \pm 0.026$ | $0.426 \pm 0.051$ |
| ROTE (severe) | $0.440 \pm 0.029$ | – | $\mathbf{0.510 \pm 0.072}$ |
| ROTE (two-stage) | $0.160 \pm 0.021$ | $0.114 \pm 0.055$ | $0.112 \pm 0.034$ |

Table 3: Single-step LLM results (with standard error) for LLM Agent Gameplay Data Prediction in *Partnr*.

In Table 4, we explored how well ROTE scales when paired with a powerful foundation model, GPT-4o. Due to cost constraints, we only compared ROTE to the most successful baseline from our prior results, NLLM. We find that across the board, ROTE outperforms NLLM even with this more powerful model. However, we observe this benefit degrades slightly in the *Partnr* benchmark, indicating the advantages of predicting in code compared to natural language may diminish in certain goal-directed, embodied settings. In Table 5, we

| Algorithm | Construction | Human | Partnr |
|---|---|---|---|
| NLLM | $0.313 \pm 0.064$ | $0.149 \pm 0.017$ | $0.78 \pm 0.042$ |
| ROTE | $\mathbf{0.566 \pm 0.028}$ | $\mathbf{0.402 \pm 0.017}$ | $0.857 \pm 0.142$ |

Table 4: Accuracy results (with standard error) across 3 datasets with models using GPT-4o as the underlying LLM.

ran a similar analysis on the Qwen models, and found that while all baselines improve with greater model size, ROTE offers a substantial boost in performance in almost all domains except for *Partnr*.

Table 5: Qwen Model Accuracy (%) and Std. Error on All Tasks and Baselines

| Model | Construction | Human | Partnr |
|---|---|---|---|
| **ROTE Baseline** | | | |
| $14B - Instruct$ | $63.78\% \pm 4.55\%$ | $50.00\% \pm 6.32\%$ | $50.00\% \pm 13.87\%$ |
| $7B - Instruct$ | $48.89\% \pm 9.14\%$ | $31.46\% \pm 2.27\%$ | $22.58\% \pm 7.63\%$ |
| $3B - Instruct$ | $45.62\% \pm 5.97\%$ | $32.50\% \pm 2.50\%$ | $26.32\% \pm 10.38\%$ |
| **NLLM Baseline** | | | |
| $14B - Instruct$ | $33.48\% \pm 4.89\%$ | $17.00\% \pm 1.51\%$ | $71.00\% \pm 4.56\%$ |
| $7B - Instruct$ | $32.21\% \pm 4.42\%$ | $12.00\% \pm 1.18\%$ | $26.00\% \pm 4.41\%$ |
| $3B - Instruct$ | $25.30\% \pm 3.87\%$ | $8.00\% \pm 0.91\%$ | $29.00\% \pm 4.56\%$ |
| **Chain-of-Thought Baseline** | | | |
| $14B - Instruct$ | $26.74\% \pm 2.96\%$ | $11.50\% \pm 0.69\%$ | $42.00\% \pm 4.96\%$ |
| $7B - Instruct$ | $24.52\% \pm 1.65\%$ | $8.60\% \pm 0.90\%$ | $12.00\% \pm 3.27\%$ |
| $3B - Instruct$ | $19.80\% \pm 3.21\%$ | $4.80\% \pm 0.66\%$ | $11.00\% \pm 3.14\%$ |
| **AutoToM Baseline** | | | |
| $14B - Instruct$ | $0.00\% \pm 0.00\%$ | $0.00\% \pm 0.00\%$ | $0.00\% \pm 0.00\%$ |
| $7B - Instruct$ | $0.00\% \pm 0.00\%$ | $0.00\% \pm 0.00\%$ | $0.00\% \pm 0.00\%$ |
| $3B - Instruct$ | $0.00\% \pm 0.00\%$ | $0.00\% \pm 0.00\%$ | $0.00\% \pm 0.00\%$ |

**Speed Analysis.** We conducted a per-model speed analysis. All models ran on a single GPU-L40. We find that across model families and sizes there is no consistent relationship between inference speed and model size on our hardware. Llama-3.1-8b-Instruct took 49 seconds to generate a complete set of 30 hypotheses, whereas the 16b Deepseek V2 Lite took 33 seconds. Interestingly, the Deepseek-Coder-V2-Lite-Instruct, which is also 16b, took 16 seconds to generate the set of hypotheses and also performed the best. The Qwen family had inference time monotonically increase with model size: Qwen2.5-3b-Instruct took 25 seconds, Qwen2.5-7b-Instruct took 43 seconds, and Qwen2.5-14b-Instruct took 79 seconds. We use the consistent scaling of the Qwen family as evidence that variations in model architecture, which is beyond our control, is the primary contributing factor to the variations in speed. As such, we use program length (in terms of number of characters) as a proxy for model efficiency. However, we note this is an imperfect assumption, since a large body of literature on topics like speculative decoding and meta-RL look for ways to distill the capabilities of large, powerful LLMs into lightweight, real-time models, making the efficiency gain of our method more nuanced.

## A.8 ROTE IN THE CONTEXT OF EXISTING PROGRAM INDUCTION METHODS

We also draw parallels to concurrent work that leverages large language models (LLMs) for program synthesis in cognitive modeling, such as CogFunSearch (Castro et al., 2025). CogFunSearch focuses on the mechanistic discovery of symbolic cognitive learning and

decision-making algorithms (such as Q-learning with forgetting terms) in dynamic multi-armed bandit tasks, operating on large datasets across multiple species. Its methodology employs a high-cost, bilevel optimization, featuring a time-intensive outer evolutionary loop to explore novel program structures ($\phi$) and an inner differentiable loop to fit continuous parameters ($\theta$). This high computational budget is justified by the complexity of simultaneously discovering program structure and fitting continuous parameters to capture subtle learning dynamics. In contrast, ROTE is engineered for the real-time action prediction problem in non-Markovian embodied settings, prioritizing scenarios where data is sparse and rapid inference is essential. ROTE eschews the evolutionary loop and continuous parameter optimization, instead relying on an efficient, single-step generative process where the LLM synthesizes a constrained space of executable program hypotheses, often implicitly modeling a Finite State Machine, from sparse observations. This results in an executable representation that enables orders-of-magnitude faster long-horizon prediction by executing inferred code directly, bypassing repeated LLM calls. While CogFunSearch excels at high-fidelity mechanistic discovery with high computational costs, ROTE offers a complementary, computationally efficient framework for representing and rapidly inferring the sequential, script-like behavioral structures prevalent in robotics and social prediction. A potential synthesis lies in using ROTE's efficiency to rapidly converge on a high-level program/script, which can then be refined using CogFunSearch's methods to tune continuous cognitive parameters within that specific structure.

Temporal Point Processes (TPPs), particularly those enhanced with logic rules, are a related research thread for modeling behavior by predicting both future action time and type based on constrained, human-readable logic (Cao et al., 2024). TPP methods like the Neuro-Symbolic TPP (NS-TPP) excel at utilizing continuous-time models and differentiable rule induction to maximize data likelihood, offering a highly precise view of event dynamics (Yang et al., 2024). ROTE, however, offers distinct advantages rooted in its executable representation. ROTE's core strength is inferring a complete, explicit behavioral program (code), which directly serves as the agent's policy for long-horizon prediction. This programmatic approach inherently provides a causal model of the agent's decision-making logic, offering greater interpretability in understanding why an action sequence occurs. While TPPs are naturally constrained by a predefined set of logical predicates, which limits their expressive range, ROTE uses a Turing-complete language (Python). This design choice enables ROTE to capture arbitrarily complex, non-Markovian behavior, making it more expressive for open-ended, embodied domains like Partnr compared to predicate-based TPPs. This difference in representation highlights their complementary focuses: TPPs are effective at predicting when the next discrete event will occur, while ROTE focuses on inferring what the agent is doing (the behavioral script).

In Table 6, we baseline against a program induction method "Iterated Hypothesis Refinement," which tries to extract rules underlying observed behavior and apply them to novel observations (Qiu et al., 2024). While our results are still preliminary, we find that this method on its own is insufficient for making robust behavioral predictions.

## A.9 Human Experiment Details

As described in Section 4, we conducted three separate human experiments: the first was collecting human gameplay data, the second was having humans predict human behavioral data, and the third was having humans predict scripted FSM agent behavior. We will open-source all of the code and stimuli used for conducting all three human experiments. For the gameplay collection, we gave participants a tutorial stage to learn the controls, and randomized the order of the tasks they played to control for ordering effects. For the behavior prediction experiments, the setup was virtually identical to that of the AI, albeit with two small modifications. The first is that we only had humans predict five timesteps into the future. This was done to make the experiment flow smoother and take less time so that participants did not fatigue for later scripts, resulting in lower prediction quality. The second change we made was we showed people 3 distinct trajectories generated by the observed agent before giving them $h_{20} = \{(o_1, a_1), (o_2, a_2) \ldots, (o_{19}, a_{19}), o_{20}\}$ and having them predict an agent's behavior. This additional context was used to help participants familiarize themselves with the dynamics of the gridworld and the space of potential agent

Table 6: Model Accuracy (%) and Std. Error comparison between ROTE and Iterated Hypothesis Refinement

| Model | Construction | Human |
|---|---|---|
| **ROTE Baseline** | | |
| DeepSeek-Coder-V2-Lite-Instruct (16B) | $52.2\% \pm 4.6\%$ | $32.9\% \pm 3.1\%$ |
| DeepSeek-V2-Lite (16B) | $31.2\% \pm 3.2\%$ | $23\% \pm 1.8\%$ |
| Llama3.1-8b-Instruct | $47.7\% \pm 4.4\%$ | $32.7\% \pm 2.6\%$ |
| Qwen-7B-Instruct | $48.89\% \pm 9.14\%$ | $31.46\% \pm 2.27\%$ |
| **Iterated Hypothesis Refinement Baseline** | | |
| DeepSeek-Coder-V2-Lite-Instruct (16B) | 7% | 4% |
| DeepSeek-V2-Lite (16B) | 19% | 0.2% |
| Llama3.1-8b-Instruct | 8% | 6.8% |
| Qwen-7B-Instruct | 7.6% | 2.8% |

behaviors. In contrast, all of our baselines only saw the current trajectory $h_{20}$. While this was done due to the limited context window of the models we used, we feel that this is still a fair comparison between humans and our baselines, since the training corpora for LLMs is rich with gridworld implementations and agent programs, and the BC model had an extended training period with the agent behavior it is predicting. In future work, we plan on relaxing this constraint by exploring dynamically growing libraries of agent programs which persist across multiple context windows, similar to an approach used in (Tang et al., 2024).

### A.10 BEHAVIOR CLONING MODEL IMPLEMENTATION DETAILS

We use an architecture and training methadology similar to the one in (Rabinowitz et al., 2018) for training a BC model with recurrence. The model uses a 2-layer ResNet to extract features from the input observations. Each observation is an image of size $70\times70$ pixels. The ResNet consists of two ResNet blocks, each containing two convolutional layers with batch normalization and a ReLU activation function. The first block uses a feature size of 64 while the second uses a feature size of 32. All blocks use stride length of 1 for all convolutional layersand a kernel size of 3.

The features extracted by the ResNet are then passed through a recurrent neural network. The model uses an LSTM with a hidden size of 128. The output of the LSTM is processed by several fully connected layers with ReLU activations. The final output is passed through a softmax layer to produce a probability distribution over the possible actions. This probability distribution represents the model's prediction of the next action an agent will take. The action space has a size of 6, corresponding to a set of discrete actions. The entire network is designed to be fully differentiable, allowing for end-to-end training using cross-entropy as the loss-function. We use the following hyperparameters for training:

| Hyperparameter | Purpose | Value |
|---|---|---|
| # Agents to Sample | The number of agent scripts to sample per epoch. | 1 |
| # Datapoints per Agent | The number of trajectories per agent to sample from the dataset per epoch. | 3 |
| # Agents | The total number of agents in the dataset. | 10 |
| # Steps | The number of steps per trajectory in the dataset. | 50 |
| Environment Size | The size of the environment. | $10\times10$ |
| Image Size | The size of a single observation in a trajectory. | $70\times70$ pixels |
| Num Epochs | The number of training epochs. | 5000 |

---

**Algorithm 1** ROTE (Representing Others' Trajectories as Executables)

---

**Require:** Observed history $h_{0:t-1} = \{(o_0, a_0), \ldots, (o_{t-1}, a_{t-1})\}$, current observation $o_t$, Environment $\mathcal{E}$, Initial set of candidate programs $\Lambda_{\text{candidates}}$ (can be empty), Initial set of program priors $P_{\text{priors}}$.
**Ensure:** Predicted action $\hat{a}_t$, Predicted programs $\Lambda_{\text{candidates}}$, Predicted program posterior $P_{\text{posteriors}}$
1: **procedure** PREDICTACTION($h_{0:t-1}, o_t, \mathcal{E}, k, \Lambda_{\text{candidates}}, P_{\text{priors}}$)
2:     **for** $N - |\Lambda_{\text{candidates}}|$ generations **do**         ▷ Number of programs to sample
3:         Prompt LLM with $h_{0:t-1}$, $o_t$, $\mathcal{E}$, and synthesize an FSM-like Python program $\lambda$
4:         $\Lambda_{\text{candidates}} \leftarrow \Lambda_{\text{candidates}} \cup \{\lambda\}$
5:         $p_{\text{prior}}(\lambda) \leftarrow \Pi_{n=1}^{|\lambda|} p_{\text{LLM}}(\text{token}_n | h_{0:t-1}, o_t, \mathcal{E}, \text{token}_{n-1}, \cdots, \text{token}_1)$
6:         $P_{\text{priors}} \leftarrow P_{\text{priors}} \cup \{p_{\text{prior}}(\lambda)\}$
7:     **end for**
8:     $P_{\text{priors}} \leftarrow \text{normalize}(P_{\text{priors}})$         ▷ Renormalize priors to account for new hypotheses
9:     $P_{\text{posteriors}} = \emptyset$
10:    **for** $\lambda \in \Lambda_{\text{candidates}}$ **do**
11:        $p(\lambda) \propto \Pi_{o_i, a_i \in h_{0:t-1}} p(a_i | o_i, \lambda) \cdot p_{\text{prior}}(\lambda)$     ▷ Calculate likelihood $p(\mathcal{H}_{[0,t-1]} | \lambda)$
12:        $P_{\text{posteriors}} \leftarrow P_{\text{posteriors}} \cup \{p(\lambda)\}$
13:    **end for**
14:    $P_{\text{posteriors}} \leftarrow \text{normalize}(\text{top-k}(P_{\text{posteriors}}, k))$        ▷ Subsample and Renormalize
15:    Predicted action $\hat{a}_t \leftarrow \text{argmax}_{a \in \mathcal{A}} \sum_{\lambda \in \Lambda_{\text{candidates}}} p_{\text{posteriors}}(\lambda) \cdot \lambda(a | o_t)$
       **return** $\hat{a}_t, \Lambda_{\text{candidates}}, P_{\text{posteriors}}$
16: **end procedure**

---

## A.11 ROTE IMPLEMENTATION DETAILS

We will fully open-source our code, including the prompts we used for generating programs with ROTE across the various levels of structure. In Algorithm 1, we show the full algorithm for ROTE and subsequently discuss the implementation details. Our approach, ROTE, constructs the program space $\Lambda$ using LLMs to synthesize executable Python programs, which serve as agent representations. These programs are structured as a class with a required act(self, observation) -> int method, ensuring a standard and executable format for all candidates. We intentionally avoid rigid, manually defined syntactic constraints across all domains to maintain representational flexibility, which is particularly important when modeling noisy human behaviors, and we analyze how these representations impact performance in Figure 13. Instead, we impose a soft constraint based on Solomonoff's theory of inductive inference and Occam's razor, encouraging the LLM to generate concise and efficient programs (minimizing $|\lambda|$), which empirically correlates with higher prediction accuracy. For a practical upper bound on program complexity, we limit the LLM's output to a maximum of 2000 tokens and restrict the number of generated hypotheses to $N = 30$, directly limiting the size of $\Lambda$ and thus the potential complexity of any individual program. Regarding the state complexity for programs that do not strictly adhere to the Finite State Machine (FSM) structure—which we permit, especially under the "Light" and "Moderate" structural conditions to accommodate inherent behavioral variability —the theoretical upper bound on the number of internal states, $|\mathcal{S}|$, is equivalent to the size of the observation space, $|\mathcal{O}|$. This is because Python is a Turing-complete language, meaning a synthesized program could, in theory, generate a unique action for every possible observation, resulting in a number of states equal to the number of unique observation-action mappings, $|\mathcal{S}| = |\mathcal{O}|$. However, as you noted, this empirical realization is highly unlikely, especially given our imposed constraint on the maximum program size. Finally, we provide control over the complexity via three levels of structural enforcement: "Light" (minimal constraint), "Moderate" (providing FSM examples), and "Severe" (enforcing a two-stage FSM generation process), allowing researchers to explore the trade-off between structure and flexibility based on the domain.

### A.11.1 ROTE HYPERPARAMETERS FOR CONSTRUCTION AND PARTNR

Across the prediction tasks for ground-truth scripted agents and humans in *Construction*, and LLM agents in *Partnr*, we used the same set of hyperparameters, indicating the generality of our method with minimal environment-specific finetuning. The only hyperparameter which varied across environments was the use of two-stage observation parsing. We used two-stage observation parsing for predicting scripted agent behavior in *Construction* and LLM-agent behavior in *Partnr*. We did not use it for predicting human behavior. As mentioned in

Section A.7, all hyperparameters were fit by comparing their performance on 20% of the data, then utilizing the best performing hyperparameter from that subset, as the selected model configuration for the remaining 80% of the data.

| Hyperparameter | Purpose | Value |
|---|---|---|
| Structure Enforcement | How strictly we constrain generated programs to adhere to FSM structure | Light |
| Rejuvenation | Whether to use rejuvenation for the FSM model. | True |
| Max rejuvenation attempts | Maximum number of times to resample a program during rejuvenation. | 2 |
| Rejuvenation threshold | The minimum number of correct action predictions a program must make over 20 timesteps to avoid resampling. | 1 |
| Max number of retries | The number of times a hypothesis can be revised if it fails to compile. | 2 |
| Number of hypotheses | The number of hypotheses to generate for the thought trace. | 30 |
| Top K | The number of most likely hypotheses to average over. | 30 |
| Minimum hypothesis probability | The minimum probability a hypothesis can have. | 1e-6 |
| Maximum number of tokens | The maximum number of tokens the large language model can generate. | 2000 |
| Minimum action probability | The minimum probability an action can have. | 1e-8 |

For our execution speed comparisons in Figure 6, all models ran on a single Nvidia GPU-L40.

**Handling Errors in Program Generation.** Given that we are generating programs from smaller LLMs trying to adhere to a consistent Agent API, and that the observation space can be challenging to operate on, there are several cases where the LLMs generate semantically meaningful programs to describe observed behaviors that fail to compile or predict actions given an observation. As such, we explored two different methods for dealing with erroneous programs. The first was revision, where we prompted an LLM to fix the code it generated given the full error trace for a program's prediction. We also gave it the original prompt and observations. The second method was completely resampling a program given the original prompt, discarding the erroneous program completely. From preliminary tests, we found completely resampling was the more effective strategy given the LLMs we were using. Since we paired this error correction process with methods like rejuvenation, we limited the number of times we could resample or revise a program to be min(Max rejuvenation attempts, Max number of retries), shared across the rejuvenation and error corrections steps. This increased the likelihood of a good program which is executable being generated, without significantly slowing our single-step inference speed.

A.11.2   EXAMPLES OF PROGRAMS GENERATED BY ROTE

In Listings 1 and 2, we show sample agent programs inferred by ROTE for the *Construction* and *Partnr* tasks, respectively. Using the same prompts and hyperparameters for both settings, our approach can flexibly model agents as Finite State Machines when the underlying agents are following scripts (*Construction*, Listing 1) or more open-ended decision makers trying to accomplish goals such as move an item from one room to another (*Partnr*, Listing 2).

```
1
2 import numpy as np
3
```

```python
class FSMAgent:
    def __init__(self, num_agents: int, num_blocks: int, num_actions: int=6):
        self.num_agents = num_agents
        self.num_blocks = num_blocks
        self.num_actions = num_actions
        self.actions = [0, 1, 2, 3, 4, 5]  # stay, right, left, down, up, interact
        self.action_to_name = ["stay", "right", "left", "down", "up", "interact"]
        self.state = "IDLE"  # Initial state

    def act(self, observation) -> int:
        agent_id = observation['agent_id']
        agent_location = observation['agent_locations'][agent_id]
        inventory = observation['agent_inventory'][agent_id]

        if self.state == "IDLE":
            # Check if there is a block at the agent's location and we can interact with it
            for block_location in observation['block_locations']:
                if np.array_equal(block_location, agent_location):
                    if inventory == -1:
                        self.state = "INTERACT"
                        break
            else:
                # No block at the agent's location, check for possible movements
                possible_actions = []
                for action in self.actions[:-1]:  # Exclude interact
                    new_location = self.apply_action(agent_location, action)
                    if not self.is_wall(new_location, observation['wall_locations']) and not self.is_other_agent(new_location, observation['agent_locations'], agent_id):
                        possible_actions.append(action)
                if possible_actions:
                    self.state = "MOVE"
                    self.target_action = np.random.choice(possible_actions)

        if self.state == "MOVE":
            self.state = "IDLE"  # Transition back to IDLE after moving
            return self.target_action

        if self.state == "INTERACT":
            self.state = "IDLE"  # Transition back to IDLE after interacting
            return 5  # Interact action

    def apply_action(self, location, action):
        if action == 1:  # right
            return [location[0], location[1] + 1]
        elif action == 2:  # left
            return [location[0], location[1] - 1]
        elif action == 3:  # down
            return [location[0] + 1, location[1]]
        elif action == 4:  # up
            return [location[0] - 1, location[1]]
        else:
            return location  # stay

    def is_wall(self, location, wall_locations):
        for wall in wall_locations:
```

```
58              if np.array_equal(wall, location):
59                  return True
60          return False
61
62      def is_other_agent(self, location, agent_locations, agent_id):
63          for i, agent_loc in enumerate(agent_locations):
64              if i != agent_id and np.array_equal(agent_loc, location):
65                  return True
66          return False
```

Listing 1: Sample Agent Codes Inferred by ROTE for *Construction* prediction task

```
1
2  import numpy as np
3
4  class FSMAgent:
5      def __init__(self, num_agents: int=1, num_blocks: int=1):
6          self.num_agents = num_agents
7          self.num_blocks = num_blocks  # irrelevant, can ignore
8
9      def parse_scene_graph(self, observation):
10         for keys in observation['scene_graph']:
11             if keys == 'furniture':
12                 for room_name, furniture_list in observation['
    scene_graph'][keys].items():
13                     for furniture_piece in furniture_list:
14                         pass  # each furniture_piece is a string
15             if keys == 'objects':
16                 if type(observation['scene_graph'][keys]) == list and
    len(observation['scene_graph'][keys]) == 0:
17                     pass  # no objects seen
18                 else:
19                     for object, object_holder_list in observation['
    scene_graph'][keys].items():
20                         for object_holder in object_holder_list:
21                             pass # each object is either on or in an
    object holder
22         return # do whatever is most helpful here
23
24     def act(self, observation) -> int:
25         '''
26         observation is a dictionary with the following keys:
27         - tool_list: List of tools available to the agent
28         - tool_descriptions: Description of how each tool is used
29         - scene_graph: Scene graph of the environment, dictionary with
     keys
30             - "furniture" which maps to a dictionary with the keys
31                 - room description string (i.e. keys could be "
    living_room_1", "bathroom_1", etc.) that maps to list of
32                     - object_id string (i.e. table_21, chair_32, etc.)
33             - "objects" which maps to a dictionary of
34                 - object_id string (i.e. keys could be "
    plate_container_2", "vase_1" etc.) to list of
35                     - object_base string (i.e "table_14", "table_21")
36                 if type(observation['scene_graph']['objects']) == list
    , then you do not observe any objects
37         - agent_state: Dictionary mapping to
38             - string of agent id (i.e. "0") maps to string describing
    what agent is doing
39         '''
40         agent_id = list(observation['agent_state'].keys())[0]
41         agent_state = observation['agent_state'][agent_id]
42         tool_list = observation['tool_list']
43
44         if 'Explore' in tool_list:
```

```
45              tool = 'Explore'
46              target = list(observation['scene_graph']['furniture'].keys
     ())[0]
47          elif 'Pick' in tool_list and 'Standing' in agent_state:
48              tool = 'Pick'
49              targets = []
50              for key in observation['scene_graph']['objects']:
51                  if 'agent_0' in observation['scene_graph']['objects'][
     key]:
52                      targets.append(key)
53              if targets:
54                  target = targets[0]
55              else:
56                  target = None
57          elif 'Place' in tool_list and 'Standing' in agent_state:
58              tool = 'Place'
59              target = None
60              for key in observation['scene_graph']['objects']:
61                  if agent_id in observation['scene_graph']['objects'][
     key]:
62                      target = key
63                      break
64              if not target:
65                  for key in observation['scene_graph']['furniture']:
66                      for furniture_piece in observation['scene_graph'][
     'furniture'][key]:
67                          if agent_id in observation['scene_graph']['
     furniture'][key]:
68                              target = key
69                              break
70              if not target:
71                  target = list(observation['scene_graph']['objects'].
     keys())[0]
72          else:
73              tool = 'Wait'
74              target = None
75
76          ## DON'T CHANGE ANYTHING BELOW HERE
77          return (tool, target, None)
```

Listing 2: Sample Agent Codes Inferred by ROTE for *Partnr* prediction task

### A.11.3 EXAMPLES OF HIGH LEVEL TRAJECTORY SUMMARIES GENERATED BY ROTE

In Listings 3 and 4, we show sample high-level trajectory summarizations from the optional two-stage observation parsing step. While in 3 the model attributes the movements of the ground truth patrolling agent as "exploring randomly," it still is able to capture some aspects of its movement, such as not interacting with blocks. In 4, ROTE can better summarize the behavior of agents in *Partnr*, but without a clear guess as to which objects the agent is trying to rearrange, it can be difficult to make a program which concisely narrows down the hypothesis space.

```
1
2 1. The agent's overall goal or strategy: The agent appears to be
     exploring its environment, possibly looking for a specific block
     or blocks.
3    It is not actively engaging with the environment in a goal-directed
      way, as it does not seem to be collecting, storing, or moving
     blocks in a strategic manner.
4
5 2. How the agent responds to different environmental features (blocks,
      walls): The agent moves around the environment, avoiding walls
     and seemingly indifferent to blocks.
```

```
 6      It repeatedly moves left and right and up and down, indicating a
          lack of strategy or goal-directed behavior.
 7
 8  3. Any patterns in movement or interaction: The agent moves in a
          pattern that suggests exploration but does not show any indication
           of avoiding walls or blocks,
 9      indicating a lack of awareness of its environment or purpose in the
           grid world.
10
11  The agent's behavior is essentially random exploration, with no
          apparent strategy or goal-directed behavior.
```

Listing 3: Sample Trajectory Summary Generated by ROTE for *Construction* prediction task

```
 1
 2  1. The agent's overall goal or strategy: The agent's main goal seems
          to be to rearrange objects in the environment, specifically
          placing them on different surfaces according to its actions. The
          actions are organized into sequences, each sequence aiming to move
           an object from its initial position to a target position.
 3  2. The agent responds to different environmental features: The agent
          interacts with specific furniture and objects within the
          environment, responding to their availability and positions. For
          example, it navigates to the kitchen to interact with a chair and
          rearrange an object on it.
 4  3. Any patterns in movement or interaction: The agent tends to move
          objects from one place to another, often towards the garage or
          kitchen, and interacts with objects like chairs and benches. It
          also arranges objects on benches and other surfaces when not
          carrying them.
 5
 6  The agent's behavior pattern emerges from a sequence of actions that
          involve picking up and moving objects to rearrange them according
          to its strategy.
```

Listing 4: Sample Trajectory Summary Generated by ROTE for *Partnr* prediction task

### A.11.4 EXAMPLES OF EXPLICIT FSMS GENERATED BY ROTE IN THE "SEVERE" STRUCTURE CONDITION

In Listings 6 and 5, we show sample FSMs predicted by ROTE which describe observed behaviors in the *Construction* and *Parntr* environments, respectively. These are produced by an LLM as an intermediate step during the "Severe" structure condition, where ROTE tries to generate agent programs implementing these high-level behavioral abstractions. As indicated by Listing 6, sometimes the models will hallucinate internal states and actions which are not within an agent's capabilities (i.e. introducing a "charge" state). As a result, the "Severe" condition is not always beneficial for model performance, especially when predicting the behaviors of goal-directed agents in partially observable settings (Figure 13).

```python
 1
 2  ```python
 3  class AgentFSM:
 4      def __init__(self):
 5          self.state = 'explore'
 6
 7      def transition(self, state, action, info):
 8          if state == 'explore':
 9              if action == 'up':
10                  return 'move_up', info
11              elif action == 'down':
12                  return 'move_down', info
13              elif action == 'left':
```

```
14                  return 'move_left', info
15              elif action == 'right':
16                  return 'move_right', info
17              elif action == 'interact':
18                  return 'interact', info
19              else:
20                  return 'stay', info
21          elif state == 'move_up':
22              return 'stay', info
23          elif state == 'move_down':
24              return 'stay', info
25          elif state == 'move_left':
26              return 'stay', info
27          elif state == 'move_right':
28              return 'stay', info
29          elif state == 'interact':
30              return 'stay', info
31
32      def get_action(self, info):
33          if self.state == 'explore':
34              # Randomly choose an action from the available options
35              import random
36              actions = ['up', 'down', 'left', 'right', 'interact']
37              action = random.choice(actions)
38              self.state, _ = self.transition('explore', action, info)
39              return action
40          else:
41              action = self.state
42              self.state, _ = self.transition(self.state, action, info)
43              return action
44  ```
45  ```
46
47  Let's write the code for the agent's behavior, taking into account the
        experiences provided:
48
49  1. The agent's overall goal or strategy: The agent appears to be
         exploring its environment, possibly looking for a specific block
         or blocks.
50      It is not actively engaging with the environment in a goal-directed
          way, as it does not seem to be collecting, storing, or moving
         blocks in a strategic manner.
51
52  2. How the agent responds to different environmental features (blocks,
         walls): The agent moves around the environment, avoiding walls
         and seemingly indifferent to blocks.
53      It repeatedly moves left and right and up and down, indicating a
         lack of
```

Listing 5: Sample FSM Transition Logic Generated by ROTE for *Construction* prediction task ("Severe" Structure Condition)

```
1  To model the behavior of the agent in this environment, we can define
       a finite state machine (FSM) with the following states and
       transitions:
2
3  **States:**
4  1. **IDLE**: The agent is waiting or resting, possibly exploring its
       surroundings to identify potential tasks or resources.
5  2. **SEARCH**: The agent is actively searching for specific objects or
        locations of interest, such as a target object to collect or a
       specific location to navigate to.
6  3. **COLLECT**: The agent is moving towards and collecting the target
       object.
```

```
 7  4. **TRANSIT**: The agent is on its way to a designated drop-off or
        storage location after collecting an object.
 8  5. **DROP_OFF**: The agent is depositing the collected object at its
        destination.
 9  6. **CHARGE**: If the agent is a robot or uses a battery, it may need
        to recharge. This state is triggered when the battery level
        becomes critical.
10
11  **Transitions:**
12  - **IDLE -> SEARCH**: When the agent identifies a task or a resource
        to collect, it transitions from an idle state to a search state.
13  - **SEARCH -> COLLECT**: When the agent locates the target object, it
        transitions from a search state to a collect state.
14  - **COLLECT -> TRANSIT**: After collecting the object, the agent
        transitions to a transit state to move towards the drop-off
        location.
15  - **TRANSIT -> DROP_OFF**: Upon reaching the drop-off location, the
        agent transitions to a drop-off state to deposit the object.
16  - **DROP_OFF -> IDLE**: After depositing the object, the agent returns
         to an idle state, possibly searching for a new task or resource.
17  - **COLLECT -> CHARGE**: If the agent is battery-operated and the
        battery level becomes too low during collection, it transitions to
         a charge state to recharge.
18  - **TRANSIT -> CHARGE**: Similarly, if the agent needs to recharge
        while moving to the drop-off location, it transitions to the
        charge state.
19  - **DROP_OFF -> CHARGE**: If the agent needs to recharge after
        depositing an object, it transitions to the charge state.
20
21  This FSM design allows the agent to efficiently manage its activities,
         transitioning smoothly between states based on its observations
        and needs, such as searching for resources, collecting them,
        moving to a drop-off location, and recharging when necessary.
```

Listing 6: Sample FSM Transition Logic Generated by ROTE for *Partnr* prediction task ("Severe" Structure Condition)

