# OpenReview forum: "Modeling Others' Minds as Code"
_ICLR.cc/2026/Conference — ICLR 2026 Poster_

### Official Review · Reviewer_G1C7 · 2025-10-18

**Soundness:** 3
**Presentation:** 3
**Contribution:** 2
**Rating:** 4
**Confidence:** 3

**Summary:**

This paper proposes ROTE, which is an algorithm that leverage LLMs as code synthesis tools to predict others’ actions. It prompt LLMs to generate computer programs explaining observed behavioral traces, then perform Bayesian inference to reason about which programs are most likely. This gives a dynamic representation that can be analyzed, modified, and composed across agents and environments. The experiments show that ROTE can predicts human and AI behaviors from sparse observations. And ROTE outperform competitive baselines including behavior cloning and LLM-based methods.

**Strengths:**

1. The Related Work section is comprehensive and well-organized. It effectively situates the current work within the broader research landscape by thoroughly reviewing and analyzing three key areas: action prediction, the use of LLMs for behavior modeling, and program induction. The discussion clearly outlines the developments and current state of these fields, providing a solid foundation for positioning the paper's contributions.
2. The proposed ROTE algorithm presents a novel approach by synergistically combining LLMs with probabilistic inference to translate observed agent behaviors into executable code programs. This represents a meaningful intermediate path between two conventional paradigms: computationally intensive inverse planning over beliefs and goals, and data-inefficient behavior cloning. By conceptualizing routines as executable "scripts" or "programs," ROTE offers a fresh perspective on modeling structured, yet flexible, agent behavior.
3. A comprehensive empirical evaluation is provided, systematically assessing performance across diverse scenarios—from scripted agents in gridworlds to complex human behaviors in embodied settings. Notably, ROTE demonstrates compelling zero-shot generalization capabilities across unseen environments, a critical feature for real-world applicability and a key advantage over more brittle baselines.

**Weaknesses:**

1. The "Related Work" section omits discussion of one relevant research thread: the use of Temporal Point Processes (TPPs) for modeling human behavior with the application of first-order logic rules to constrain behavior prediction. These areas are closely related to the methodological context of this paper.
2. The central claim of modeling behavior as "programs" lacks sufficient discussion of their connection to underlying cognitive processes. A key theoretical concern is whether finite-state machines can fully capture the richness of human behavior, particularly for goal-directed or socially complex scenarios. This fundamental limitation warrants deeper theoretical grounding, as it directly challenges the generalizability of the proposed approach.
3. The experimental evaluation has several limitations. While the paper highlights ROTE's long-term prediction efficiency, it lacks comprehensive discussion of computational costs—particularly the potentially significant overhead during initial program synthesis, which could impact real-time applicability.

**Questions:**

1. It lacks a clear operational definition of "scripted" behavior, leaving the boundary between scripted and non-scripted behaviors ambiguous. This conceptual vagueness—such as whether goal-directed planning qualifies as scripted—undermines the theoretical grounding of the proposed approach. Additionally, while ROTE demonstrates strong predictive accuracy, the semantic validity and interpretability of the generated programs remain underexplored. A more systematic analysis is needed to assess whether these programs meaningfully correspond to plausible behavioral logic or merely capture superficial statistical patterns.
2. The construction of the program space $\Lambda$ — including its syntactic constraints and upper bounds on state complexity — remains insufficiently specified. A more detailed description of these structural parameters would significantly enhance the reproducibility and rigorous evaluation of the proposed method.
3. The core modeling assumption of combining deterministic programs with a noise model appears overly simplistic. Given the inherently stochastic and context-dependent nature of genuine human behavior, merely approximating behavioral uncertainty through a noise model may be insufficient to capture its full complexity, potentially limiting the model's realism and generalizability.
4. What is the requirement for different levels of structural constraints?
5. While multi-step prediction is efficient, the computational overhead during the initial program synthesis phase has not been compared against baseline methods. Could the authors provides more comparative measurements of this initial synthesis stage?
6. The paper does not analyze how LLM scale affects performance. While multiple LLMs are used, their varying sizes' impact on program quality and inference speed remains unexplored.
7. The TPP methods, such as [1, 2]，with the constraint of human-readable logic ruels, can be used to predict both future action time and type. Compared with these methods, what are the advantages of ROTE? Is it possible to add extra experiments to compare?

[1] Neuro-Symbolic Temporal Point Processes. ICML 2024

[2] Discovering Intrinsic Spatial-Temporal Logic Rules to Explain Human Actions. NeurIPS

---

> ### Author Response · Authors · 2025-11-24
>
> - We appreciate the reviewer’s questions about the relationship between model size and prediction quality. Following your suggestions, we ran new experiments which analyzed more model sizes’ predictive accuracy on all of the evaluation conditions. Due to cost and time constraints, we only ran the most successful baselines (NLLM, as you suggested) and our method, ROTE, with GPT-4o. We find that NLLM using GPT-4o gets 31.33%, 14.93%, and 78.00% accuracy on the Construction (multi-step), Human (multi-step), and Partnr datasets respectively. In contrast, ROTE using GPT-4o got accuracies of 56.57%, 40.21%, and 85.71% on the Construction (multi-step), Human (multi-step), and Partnr datasets respectively. This indicates that even with more powerful models, our approach offers a significant boost in performance, while remaining more cost effective since the generated programs can be reused across contexts rather than having to reprompt the LLM for every prediction. This also helps us be more confident that our evaluation settings pose challenging problems, as if the success in behavioral predictions for ROTE were due to picking up on simple statistical noise, we might expect the benefit to be less noticeable as model size increases. We also tried a different model family: Qwen2.5-3B-Instruct, Qwen2.5-7B-Instruct, Qwen2.5-14B-Instruct. We again find that across all model sizes and evaluation settings, ROTE offers a substantial gain that scales with model size. Due to the number of results, we will report the full table in the appendix of the paper (Tables 1, 2, 3 and 5). For illustration, however, we find that the 3B/7B/14B accuracy for NLLM using Qwen2.5 on the Construction dataset is 25.30%, 32.21%, and 33.48% respectively, whereas ROTE had an accuracy of 45.62%, 48.89%, and 63.78% respectively. We again thank the reviewer for this suggestion as we believe it strengthens the empirical validity of our method. We will be sure to include all of the results in a new draft of the paper, as well as evaluations on larger models from different families.  We also conducted a per-model speed analysis. All models ran on a single GPU-L40. We find that across model families and sizes there is no consistent relationship between inference speed and model size on our hardware. Llama-3.1-8b-Instruct took 49 seconds to generate a complete set of 30 hypotheses, whereas the 16b Deepseek V2 Lite took 33 seconds. Interestingly, the Deepseek-Coder-V2-Lite-Instruct, which is also 16b, took 16 seconds to generate the set of hypotheses and also performed the best. The Qwen family had inference time monotonically increase with model size: Qwen2.5-3b-Instruct took 25 seconds, Qwen2.5-7b-Instruct took 43 seconds, and Qwen2.5-14b-Instruct took 79 seconds. We use the consistent scaling of the Qwen family as evidence that variations in model architecture, which is beyond our control, is the primary contributing factor to the variations in speed. As such, we use program length (in terms of number of characters) as a proxy for model efficiency. That said, we will revise the writing of the paper to acknowledge that this is an imperfect assumption, since a large body of literature on topics like speculative decoding and meta-RL look for ways to distill the capabilities of large, powerful LLMs into lightweight, real-time models, making the efficiency gain of our method more nuanced.
>
> - We thank the reviewer for the insightful comment about how our algorithm’s impact could be improved by extending to stochastic behavior inference as well. By no means do we argue that Finite State Machines are the only model one may need (our results in Figure 13 certainly support that in complex settings, more open-ended code is needed). In many situations humans are noisy, and right now our method tries to account for this with a noise model. We feel the limitation stems from practical constraints rather than algorithmic ones: initially we tried to have LLMs attempt to generate probabilistic predictions about future actions, but they often gave poor confidence estimates when predicting this as code. ROTE currently uses a deterministic policy for efficiency, handling stochasticity via a fixed ϵ noise model, but we agree this is insufficient for robust modeling. Our algorithm does not make any strong commitments to how the probabilities of the actions are generated (deterministically with a noise model or through a probabilistic generative model). As such, our future research will focus on having ROTE generate probabilistic programs to offload uncertainty calculations to a verified external library like [memo](https://github.com/kach/memo) (a tool ideal for belief inference using JAX's speed). This preserves ROTE's efficiency while enabling rigorous reasoning about internal states with stochastic transitions between them. The main challenge is currently the scarcity of training data for LLMs to generate correct probabilistic social reasoning code.

---

> > ### Author Response · Authors · 2025-11-24
> >
> > - We appreciate the reviewer's query regarding the computational overhead of the initial program synthesis phase. We have now conducted a direct comparison of the time taken (in seconds) and the resulting error at the first prediction timestep for all methods. The results show that the BC baseline is the fastest, requiring only 0.58 seconds with an associated error of 0.03. In contrast, our proposed method, ROTE, takes 13.34 seconds with an error of 0.42. The other comparative methods include NLLM, which takes 5.62 seconds with the lowest error of 0.02, and AutoToM, which is the slowest at 63.06 seconds with the highest error of 0.69. In Figure 6 we plot these results when (# of predictions) is 0, and show that while ROTE has a slower initial generation phase, its multistep prediction is more efficient in the long run, even when compared to BC.
> >
> > - We appreciate the reviewer’s feedback regarding the insufficient specification of the program space $\Lambda$ and its constraints, as we agree that a clear definition of these structural parameters is essential for reproducibility and rigorous evaluation of the proposed method. Our approach, ROTE, constructs the program space $\Lambda$ using LLMs to synthesize executable Python programs, which serve as agent representations. These programs are structured as a class with a required act(self, observation) -> int method, ensuring a standard and executable format for all candidates. We intentionally avoid rigid, manually defined syntactic constraints across all domains to maintain representational flexibility, which is particularly important when modeling noisy human behaviors, and we analyze how these representations impact performance in Figure 13. Instead, we impose a soft constraint based on Solomonoff's theory of inductive inference and Occam's razor, encouraging the LLM to generate concise and efficient programs (minimizing $|\lambda|$), which empirically correlates with higher prediction accuracy. For a practical upper bound on program complexity, we limit the LLM's output to a maximum of 2000 tokens and restrict the number of generated hypotheses to $N=30$, directly limiting the size of $\Lambda$ and thus the potential complexity of any individual program. Regarding the state complexity for programs that do not strictly adhere to the Finite State Machine (FSM) structure—which we permit, especially under the "Light" and "Moderate" structural conditions to accommodate inherent behavioral variability —the theoretical upper bound on the number of internal states, $|\mathcal{S}|$, is equivalent to the size of the observation space, $|\mathcal{O}|$. This is because Python is a Turing-complete language, meaning a synthesized program could, in theory, generate a unique action for every possible observation, resulting in a number of states equal to the number of unique observation-action mappings, $|\mathcal{S}| = |\mathcal{O}|$. However, as you noted, this empirical realization is highly unlikely, especially given our imposed constraint on the maximum program size. Finally, we provide control over the complexity via three levels of structural enforcement: "Light" (minimal constraint), "Moderate" (providing FSM examples), and "Severe" (enforcing a two-stage FSM generation process), allowing researchers to explore the trade-off between structure and flexibility based on the domain. We have added this explanation to section A.11 in the Appendix.
> >
> > - We thank the reviewer for giving us the opportunity to clarify our use of goal-directed behavior and how it fits within the soundness of our method. While there is a large body of literature and debate surrounding what it means to be goal-directed, in this work we say any agents conducting forward plans, denoted by explicit rollouts within the environment, are considered to be goal-directed. In the Construction gridworld task, this means the agents using A-star to complete tasks such as “pickup green blocks and move them to the corner” and “pair all blue blocks together” are goal-directed, whereas “patrol the grid in a clockwise direction,” which uses the Manhattan distance and FSM states as planning heuristics, are considered scripted. In Partnr, all of the LLM agents are goal-directed since they use the ReAct framework to plan how to complete household tasks. When collecting human data, we do not know if the participants are conducting detailed planning. We note that while we motivate our work from prior literature in cognitive science about predicting the behavior of scripted agents, our empirical results demonstrate that our approach is robust to predicting goal-directed behavior in the sense that we define it here. Our definition of goal-directed and the distinction between our motivation and the application of our algorithm is one we will try to make more clear in a revised version of the paper, and we thank the reviewer for encouraging us to be more precise in our explanations.

---

> > > ### Author Response · Authors · 2025-11-24
> > >
> > > - Would the reviewer be able to clarify what they mean by “requirement for different levels of structural constraints?” Once we understand the precise requirements you are interested in, we would be happy to provide a detailed explanation.
> > >
> > > - We appreciate the reviewer’s reference to [1] and [2]. We will cite the mentioned works in the Related Works Section and exactly add the following paragraph to Section A.8 in the Appendix : “Temporal Point Processes (TPPs), particularly those enhanced with logic rules, are a related research thread for modeling behavior by predicting both future action time and type based on constrained, human-readable logic. TPP methods like the Neuro-Symbolic TPP (NS-TPP) excel at utilizing continuous-time models and differentiable rule induction to maximize data likelihood, offering a highly precise view of event dynamics. ROTE, however, offers distinct advantages rooted in its executable representation. ROTE's core strength is inferring a complete, explicit behavioral program (code), which directly serves as the agent's policy for long-horizon prediction. This programmatic approach inherently provides a causal model of the agent's decision-making logic, offering greater interpretability in understanding why an action sequence occurs. While TPPs are naturally constrained by a predefined set of logical predicates, which limits their expressive range, ROTE uses a Turing-complete language (Python). This design choice enables ROTE to capture arbitrarily complex, non-Markovian behavior, making it more expressive for open-ended, embodied domains like Partnr compared to predicate-based TPPs. This difference in representation highlights their complementary focuses: TPPs are effective at predicting when the next discrete event will occur, while ROTE focuses on inferring what the agent is doing (the behavioral script).”

---

> ### Comment · Reviewer_G1C7 · 2025-11-27
>
> Thanks for the response and additional analysis. To clarify, my question regarding “different levels of structural constraints (last paragraph of Section 5)” concerns the following: What precisely constitutes each level of structural constraint? Under what circumstances should each level be applied, and why do different environments require different degrees of structure? Finally, how do these structural constraints affect the model’s performance and generalization?

---

> > ### Author Response · Authors · 2025-11-28
> >
> > Thank you for clarifying your question and for actively engaging in the review process. The first constraint “light” is done by prompting the LLM to generate agent code that explains the observed behavior. In our prompt, we say the agents the code is describing are structured as FSMs, but we do not give explicit information beyond that about what the logic of these FSMs will look like or possible internal states. The “moderate” constraint is done by having a similar prompt, with additional context describing potential internal states and their transition logic, such as “States:1. SEARCH - Agent searches for nearest uncollected block 2. COLLECT - Agent moves toward and collects target block 3. RETURN - Agent returns to base location to deposit block Transitions: - SEARCH -> COLLECT: When agent sees an uncollected block - COLLECT -> RETURN: When agent has collected block in inventory…” The third constraint “severe” is similar to moderate, except after giving the LLM examples of states and transition logic, we have it explicitly generate hypothesized states and transitions in natural language before being asked to implement that hypothesis in code. In our results, which we discuss in the final paragraph of section 5, Appendix A.4, and Figure 13, we find that the optimal level of structural constraint is highly dependent on the environment and the nature of the agent's behavior. For scripted agents in the fully observable Construction environment (following deterministic Finite State Machines), the "Severe" approach performed as well as others, suggesting that for predictable, rote behaviors, a strictly structured FSM representation is effective and computationally efficient. This structure serves as a useful inductive bias for agents interacting with simple, rule-based systems. However, when modeling human behavior in Construction, the "Moderate" condition was superior. This highlights the need for representational flexibility when agents follow a general script but exhibit inherent variability. Conversely, for predicting the behavior of goal-directed agents in the partially observable Partnr environment, forcing agents into a strict FSM representation ("Severe") significantly hurt performance. This is likely because the rigidity of the FSM structure fails to account for how goal-directed agents react to the "unknown unknowns" encountered in a partially observable world. Therefore, less structure ("Light") is often better in complex, partially observable settings as it allows for a wider range of decision-making processes to be inferred. The effect of these constraints on performance can be visually seen in the provided ablation study (Figure 13). While these are our preliminary investigations into when each structure should be applied, we are excited by future work which may use meta-reinforcement learning to decide the optimal level depending on the context.

---

### Official Review · Reviewer_P8y6 · 2025-10-29

**Soundness:** 2
**Presentation:** 3
**Contribution:** 2
**Rating:** 4
**Confidence:** 4

**Summary:**

The paper presents ROTE, a novel method that models others' minds as code. It leverages LLMs to synthesize behavioral programs from sparse observations and uses probabilistic inference to predict actions. Experiments show ROTE outperforms baselines and achieves human-level accuracy in predicting both human and AI behavior.

**Strengths:**

1. The paper is generally well-written and structured.

2. The method shows performance gains over the selected baselines, and in one task, it is reported to perform at a level comparable to human predictors.

**Weaknesses:**

1. There is a lack of discussion regarding [1], which is a representative work that also utilizes LLMs to generate open-ended code for observed behaviors. While I can appreciate the potential differences, the complete absence of any discussion of this work is surprising.

2. The claimed contribution of a "novel algorithm" does not seem to hold, in my opinion (at least pending further clarification). I find the architecture to be strikingly similar to that of [2], which might even be more sample-efficient. While some differences may exist in the format of the hypotheses or the use of weights, I consider these distinctions to be trivial. The complete omission of this highly relevant work is unacceptable.

3. Given that the paper positions itself in the field of Program Induction, it should be compared with more related work from this area. My intuition is that some of these works, such as [3], could be directly applied to the proposed task (with prompt modifications, of course) and would likely achieve comparable results.

Overall, I see too many shadows of prior work in this paper, along with potential baselines that should have been discussed but were omitted. While I appreciate the paper's motivation and writing, I believe it is not yet ready for acceptance.

[1] Castro, Pablo Samuel, et al. "Discovering symbolic cognitive models from human and animal behavior." bioRxiv (2025): 2025-02.

[2] Zhou, Yangqiaoyu, et al. "Hypothesis generation with large language models." arXiv preprint arXiv:2404.04326 (2024).

[3] Qiu, Linlu, et al. "Phenomenal yet puzzling: Testing inductive reasoning capabilities of language models with hypothesis refinement." arXiv preprint arXiv:2310.08559 (2023).

**Questions:**

See Weakness.

---

> ### Author Response · Authors · 2025-11-24
>
> - We thank the reviewer for highlighting the omission of a detailed discussion of [1] (which we refer to as CogFunSearch) and acknowledge its significance as a representative work utilizing LLMs to generate open-ended code for observed behaviors. We will cite the paper in Section 2, Related Work, and exactly add the following text to the paper in Appendix A.8: “ROTE draws parallels to concurrent work that leverages large language models for program synthesis in cognitive modeling, such as CogFunSearch. CogFunSearch focuses on the mechanistic discovery of symbolic cognitive learning and decision-making algorithms (such as Q-learning with forgetting terms) in dynamic multi-armed bandit tasks, operating on large datasets across multiple species. Its methodology employs a high-cost, bilevel optimization, featuring a time-intensive outer evolutionary loop to explore novel program structures (ϕ) and an inner differentiable loop to fit continuous parameters (θ). This high computational budget is justified by the complexity of simultaneously discovering program structure and fitting continuous parameters to capture subtle learning dynamics. In contrast, ROTE is engineered for the real-time action prediction problem in non-Markovian embodied settings, prioritizing scenarios where data is sparse and rapid inference is essential. ROTE eschews the evolutionary loop and continuous parameter optimization, instead relying on an efficient, single-step generative process where the LLM synthesizes a constrained space of executable program hypotheses, often implicitly modeling a Finite State Machine (FSM), from sparse observations. This results in an executable representation that enables orders-of-magnitude faster long-horizon prediction by executing inferred code directly, bypassing repeated LLM calls. While CogFunSearch excels at high-fidelity mechanistic discovery with high computational costs, ROTE offers a complementary, computationally efficient framework for representing and rapidly inferring the sequential, script-like behavioral structures prevalent in robotics and social prediction. A potential synthesis lies in using ROTE's efficiency to rapidly converge on a high-level program/script, which can then be refined using CogFunSearch’s methods to tune continuous cognitive parameters within that specific structure.
>
> - We sincerely apologize for not including [2], (which we will refer to as HypoGeniC for clarity), within our discussion, and will include it in our related works section. We agree that the core idea of using LLMs to synthesize and refine hypotheses is a shared and powerful methodology. However, the core motivation behind ROTE is rooted in the cognitive science insight that much of human behavior follows efficient, script-like routines, a departure from HypoGeniC, which informs the key differences we highlight in representational format, algorithmic loop, and prediction capability. The representational difference is that HypoGeniC generates natural language hypotheses (e.g., "Tweets with emotional tones are retweeted more" ) about observational data, which are then used by an LLM classifier to make a prediction. ROTE, however, generates fully executable computer programs (code) that act as an agent's policy (π) and state transition function (u). The difference is important to the motivation behind our work; we believe that programs can be a useful representation because of the cognitive science insight that much of human behavior follows scripts, and because they enable inferring a causal model of decision-making logic. : HypoGeniC instead infers features that may help with  classification.Its core contribution is its iterative, data-driven reward-based mechanism for refining a natural language hypothesis bank across trials and examples. In contrast, ROTE's core novelty lies in the ability to infer a programmatic behavioral policy from sparse observations and prove that this representation efficiently generalizes and scales to long-horizon, embodied, and non-Markovian environments. This move from general natural language hypotheses to specific executable programs shifts the prediction task from one of LLM-based reasoning/classification to one of code execution, leading to dramatic differences in application and performance. Our results in Figure 13 demonstrate that the choice of representational structure, significantly impacts predictive accuracy in embodied domains. Furthermore, the executable nature of ROTE's programs provides an orders of magnitude speedup for multi-step prediction compared to any LLM-based reasoning approach (including AutoToM, which is similar to HypoGeniC and was awarded a spotlight in NeurIPS 2025), precisely because execution replaces repetitive LLM prompting. We will explicitly clarify these distinctions in the revised manuscript, dedicating a section to contrasting ROTE's program synthesis with HypoGeniC’s natural language hypothesis generation.

---

> > ### Author Response · Authors · 2025-11-24
> >
> > - We appreciate the insightful suggestion to directly compare ROTE against models from the Program Induction field, specifically addressing the capability of models like [3] to perform hypothesis refinement for our task. We directly use the code from this paper (which we will call IHR - Iterated Hypothesis Refinement), with a minor extension to support open-source models and custom prompts for our specific environments, and have already obtained preliminary results on the Construction environment for both scripted and human agents using several LLM families. The current multi-step prediction accuracy for this new baseline (with various LLMs) for IHR is significantly lower than ROTE's performance (e.g., ROTE achieves ≈47% for scripted and ≈31% for human agents in multi-step prediction). Specifically, the accuracy for IHR predicting scripted agents in Construction is currently low: Llama 3.1 reached 8%, DeepSeek V2 Lite reached 19%, DeepSeek V2 Coder reached 7%, and Qwen 2.5 7B Instruct reached 7.6%. Similarly, for human agents, the performance is 6.8%, 0.2%, 4%, and 2.8%, respectively. We are continuing to optimize this baseline with more model sizes and families, and are working to adapt it to the complexity of the Partnr environment. We will update the next draft with the complete set of final results and a detailed discussion comparing this program induction technique to ROTE. For now, tables of our work can be found in Table 6 and Appendix A.8.

---

> > > ### Comment · Reviewer_P8y6 · 2025-11-26
> > >
> > > Thank you for the response, but I would like to ask for further clarification on the following points:
> > >
> > > 1.	Regarding the comparison with HypGenic, I understand the difference in the form of the hypotheses. However, I would like to know: if the hypotheses in HypGenic were also converted into code, what are the main innovations or specific design choices in your algorithm that you would highlight as going beyond that baseline?
> > >
> > > 2.	Could you please provide more details about the implementation of IHR method? In particular, how did you set up the prompts and hyperparameters, and which samples were provided at each iteration? I would like to make sure that the comparison is conducted fairly.

---

> > > > ### Author Response · Authors · 2025-11-28
> > > >
> > > > Thank you for actively engaging with us in the review process and for the opportunity to clarify the distinctions between ROTE and HypoGeniC. ROTE's core novelty lies in its representation: we believe framing behavior prediction as the synthesis and selection of executable policies is a novel perspective on this core challenge for AI. Towards that end, we view this work as having **five key architectural and algorithmic innovations** for successful inference of embodied agents’ behavior:
> > > >
> > > > 1. **Probabilistic Inference Algorithm**: ROTE employs a Bayesian Inverse Planning mechanism adapted for executable programs as its “scoring” function of different hypotheses. We calculate the posterior probability of a program λ based on the trajectory likelihood, maximizing the fit to the entire sequence of observed actions and observations (h0:t−1​). This is possible because we ask the LLM to generate code which relies on state-based tracking (via FSM execution) of the observed agent’s history. HypoGeniC uses a non-sequential, UCB-inspired scoring function that maximizes a hypothesis's historical accuracy (average reward) over a "bag" of examples. This metric cannot account for the temporal or conditional dependencies inherent in a long-horizon behavioral trace. Our sequential inference is thus essential for modeling the causal structure of agents’ behavior.
> > > >
> > > > 2. **Hypothesis-conditioned behavior prediction mechanism**: ROTE's output is the agent's policy. Once the program is inferred and weighted, ROTE simply executes the code directly to produce multi-step predictions (a^t​,…,a^t+10​). This is an architectural choice that enables ROTE's compute cost to scale orders of magnitude more efficiently with the number of future predictions. HypoGeniC's final prediction, even when guided by a hypothesis, requires the LLM to perform text-based reasoning in a prompt for every new example. This process is inherently high-latency and costly, necessitating repeated, expensive LLM calls for each step of a multi-step prediction. Even if HypoGeniC was modified to have the hypotheses be in code, its current algorithm would have an LLM condition on this code to make predictions, whereas we undertook a significant amount of engineering effort to make sure LLMs could successfully generate programs and execute them directly.
> > > >
> > > > 3. **Prior over hypotheses**: We integrate an explicit design principle where the LLM is prompted to generate concise and efficient code. This prior is formally grounded in Solomonoff's theory of inductive inference and Occam's razor. Our empirical results confirm that this constraint yields a strong inductive bias, as shorter program length (∣λ∣) correlates directly with higher predictive accuracy and generalization. HypoGeniC's reward and update mechanisms do not encode a length or complexity penalty on the generated natural language hypotheses or anywhere in their evaluation loop.
> > > >
> > > > 4. **Adaptive Hypothesis Set Management**: ROTE uses Sequential Monte Carlo (SMC) with rejuvenation. If a program's posterior likelihood falls below a threshold (meaning it poorly explains the observed sequence), it is explicitly replaced (rejuvenated) by a newly synthesized program from the LLM. This dynamically refreshes the hypothesis pool with new, diverse possibilities to maintain relevance. HypoGeniC's mechanism relies on triggering new hypothesis generation only when a wrong example bank reaches a maximum size. While this is an effective updating mechanism, it is a purely additive process focused on covering unexplainable examples, rather than a probabilistic method for iteratively pruning and replacing poorly performing hypotheses based on their accumulated trajectory likelihood.
> > > >
> > > > 5. **Representational Choice for Embodied Behavior Prediction**: ROTE's design goal is the synthesis of executable policies to achieve superior zero-shot generalization to novel environments. Our initial explorations revealed that while generating fully probabilistic programs was challenging (due to current LLMs providing poor, often uniform, uncertainty estimates of their predictions in code), adopting a code format which produced deterministic outputs combined with a noise model was the most robust and performant path. Design decisions such as these are a significant part of our algorithm that HypoGeniC does not give an immediate insight on how to handle. HypoGeniC generates natural language rules for classification, often for text-based tasks. The required output is a high-level, human-interpretable theory rather than a low-level, self-contained execution policy that models physical agents in partially observable, embodied worlds. We are excited about future work combining the best of ROTE and HypoGeniC by generating natural language hypotheses about an agent’s beliefs and goals, then translating them into executable probabilistic programs such as [memo](https://github.com/kach/memo), a PPL specialized for social reasoning over these artifacts..

---

> > > > > ### Author Response · Authors · 2025-11-28
> > > > >
> > > > > Thank you for following up on our implementation details for the IHR method. We cloned their repo directly, and following their instructions, we converted our dataset of (state, action) pairs into the .jsonl format. We used the exact same 100 datapoint evaluation set we evaluated all of our other baselines on. Each datapoint consisted of 20 state action pairs in the “training” set, and the remaining 10 used for multi-step prediction in the “test” set (in the human evals the test set has 5 state action pairs). As a sample of one of the datapoints, we include the following:
> > > > > ```
> > > > > {"idx": 0, "train": [{"input": ["{'agent_id':6,'agent_inventory':array([-1],dtype=int32),'agent_inventory_colors':array([[-1,-1,-1]],dtype=int32),'agent_locations':array([[3,5]],dtype=int32),'block_colors':array([[0,255,0],[0,0,255],[128,0,128],[255,192,203],[0,255,255],[0,255,0]],dtype=int32),'block_locations':array([[4,4],[3,3],[1,1],[1,4],[1,5],[4,1]],dtype=int32),'terminal':False,'time':0,'wall_locations':array([[0,0],[1,0],[2,0],[3,0],[4,0],[5,0],[6,0],[0,6],[1,6],[2,6],[3,6],[4,6],[5,6],[6,6],[0,0],[0,1],[0,2],[0,3],[0,4],[0,5],[0,6],[6,0],[6,1],[6,2],[6,3],[6,4],[6,5],[6,6],[4,3],[4,2],[3,2]],dtype=int32)}"], "output": "right"}, {"input": ["{'agent_id':6,'agent_inventory':array([-1],dtype=int32),'agent_inventory_colors':array([[-1,-1,-1]],dtype=int32),'agent_locations':array([[4,5]],dtype=int32),'block_colors':array([[0,255,0],[0,0,255],[128,0,128],[255,192,203],[0,255,255],[0,255,0]],dtype=int32),'block_locations':array([[4,4],[3,3],[1,1],[1,4],[1,5],[4,1]],dtype=int32),'terminal':False,'time':1,'wall_locations':array([[0,0],[1,0],[2,0],[3,0],[4,0],[5,0],[6,0],[0,6],[1,6],[2,6],[3,6],[4,6],[5,6],[6,6],[0,0],[0,1],[0,2],[0,3],[0,4],[0,5],[0,6],[6,0],[6,1],[6,2],[6,3],[6,4],[6,5],[6,6],[4,3],[4,2],[3,2]],dtype=int32)}"], "output": "right"}, …], "test": [{"input": ["{'agent_id':6,'agent_inventory':array([-1],dtype=int32),'agent_inventory_colors':array([[-1,-1,-1]],dtype=int32),'agent_locations':array([[4,4]],dtype=int32),'block_colors':array([[0,255,0],[0,0,255],[128,0,128],[255,192,203],[0,255,255],[0,255,0]],dtype=int32),'block_locations':array([[4,4],[3,3],[1,1],[1,4],[1,5],[4,1]],dtype=int32),'terminal':False,'time':20,'wall_locations':array([[0,0],[1,0],[2,0],[3,0],[4,0],[5,0],[6,0],[0,6],[1,6],[2,6],[3,6],[4,6],[5,6],[6,6],[0,0],[0,1],[0,2],[0,3],[0,4],[0,5],[0,6],[6,0],[6,1],[6,2],[6,3],[6,4],[6,5],[6,6],[4,3],[4,2],[3,2]],dtype=int32)}"], "output": "down"},...]}
> > > > > ```
> > > > > In terms of prompts, we adapted the original authors’ prompts from the [arc](https://github.com/linlu-qiu/lm-inductive-reasoning/blob/main/prompts/arc.py) task with the following modifications:
> > > > > ```
> > > > > rule_to_python_prompt = """You are an expert Python programmer. Write a Python function `fn` for the following rule. The input is a series of labels about object locations and their respective coordinates in a 2d grid. The output is an action from the list ['stay', 'right', 'left', 'down', 'up', 'interact']. (rest of original prompt is same)"""
> > > > > rule_to_output_prompt = """Generate an output for the given input based on the following rule. The input is a series of labels about object locations and their respective coordinates in a 2d grid. The output is an action from the list ['stay', 'right', 'left', 'down', 'up', 'interact']. (rest of original prompt is same)"""
> > > > > rule_to_output_prompt_with_format = """Generate an output for the given input based on the following rule. The input is a series of labels about object locations and their respective coordinates in a 2d grid. The output is an action from the list ['stay', 'right', 'left', 'down', 'up', 'interact']. (rest of original prompt is same)"""
> > > > > ```
> > > > > For hyperparameters, we tried to keep the evaluation as close to ROTE as possible: number of hypotheses generated = 30; number of iterations (same as number of retries in our algorithm) = 2; temperature = 1.0. All other hyperparameters were kept at their default values. If the reviewer has suggestions for how we might improve this baseline to provide a more fair comparison we would be more than happy to integrate them!

---

### Official Review · Reviewer_vNqa · 2025-10-31

**Soundness:** 3
**Presentation:** 2
**Contribution:** 3
**Rating:** 6
**Confidence:** 3

**Summary:**

This paper introduces a novel approach to modeling others’ minds as code. It addresses the challenge of predicting agents’ actions in scenarios where behavior is not guided by explicit goals or beliefs, inspired by cognitive science studies on mindless behavior. The proposed ROTE algorithm integrates LLM-generated code with Bayesian inference to identify programs that best explain an agent’s observed behavior and to predict future actions. Experiments in two environments show substantial improvements over strong baselines.

**Strengths:**

- Motivated by findings in cognitive science, this work addresses a gap in machine mind modeling with potentially broad implications.

- Experiments in two distinct environments show that ROTE consistently outperforms BC, AutoToM, and a naive LLM baseline in both single-step and multi-step prediction tasks. The method also demonstrates strong computational efficiency, particularly in long-horizon tasks.

**Weaknesses:**

- The baselines and ROTE are evaluated with 8B and 16B, instruct and coder models. I wonder if more capable LLMs (such as GPT-4o) could directly solve the action prediction tasks as NLLM, without additional modeling approaches. This would provide a more realistic assessment of the practical need for such a modeling approach.

- The proposed algorithm would be more significant if the generated programs could also handle action prediction tasks that require modeling agents’ beliefs, such as the Forward Action task in BigToM.

- The description of the generalization experiments lacks some details. It would improve reproducibility to specify the environments used before and after the change. How are the textual descriptions different between the two environments?

**Questions:**

- How are different candidate programs generated, and how do you ensure their diversity? Is this achieved through a multi-pass or single-pass generation process?
- Could you evaluate stronger LLMs to better illustrate the necessity of these action prediction tasks and how current models fall short?
- How do the baselines (NLLM and AutoToM) with GPT-4o as their backend perform on the action prediction tasks compared to ROTE?

---

> ### Author Response · Authors · 2025-11-24
>
> - We appreciate the reviewer encouraging us to more deeply evaluate the relationship between model scale and the necessity of our modeling framework. Based on your suggestion, we re-ran our top-performing models (ROTE and NLLM) with GPT-4o; however, due to cost and time constraints, we were unable to run the entire suite of baselines. We find that NLLM using GPT-4o gets 31.33%, 14.93%, and 78.00% accuracy on the Construction (multi-step), Human (multi-step), and Partnr datasets respectively. In contrast, ROTE using GPT-4o got accuracies of 56.57%, 40.21%, and 85.71% on the Construction (multi-step), Human (multi-step), and Partnr datasets respectively. This indicates that even when compared against predictions from more powerful models like GPT-4o, our approach, ROTE, offers a significant boost in performance over a powerful base model, while simultaneously pushing the frontier of cost-effective and generalizable action prediction. This also helps us be more confident that our evaluation settings pose challenging problems, because if the success in behavioral predictions for ROTE were due to picking up on simple statistical noise, we might expect the benefit to be less noticeable as model size increases. We additionally ran experiments with a different model family (Qwen2.5-3/7/14 B-Instruct) and with chain-of-thought reasoning for Reviewer 5dXd. Our results corroborate previous findings by showing that across all model sizes and evaluation settings, ROTE offers a substantial gain that scales with model size. We report the full table in the appendix of the revised paper (Tables 1, 2, 3, and 5). We again thank the reviewer for this suggestion as we believe it strengthens the empirical validity of our method, and will continue to evaluate larger models and different model families.
>
> - Thank you for the opportunity for us to describe our generalization evaluation in greater detail. Our Construction gridworld environment uses an API similar to that in [JaxMARL](https://github.com/FLAIROx/JaxMARL). We write its “reset” function so that given a pseudorandom seed, the initial positions of agents, colored blocks, and walls are randomly shuffled throughout the environment. The observation spaces in text are the x,y coordinates of different objects and agents throughout the grid and the action taken from that state. I.e. we say “Wall locations: [[wall1_x, wall1_y], … [wallN_x, wallN_y]] \n Agent locations: [[agent1_x, agent1_y]] … \n Colored Block locations: … Agent Action: Left”. We show all models the state in an environment generated using a pseudorandom seed from times 0 to 20. Then, using a distinct seed, we reset the environment, which shuffles item and agent locations. We take the first observation produced by this reset and concatenate it to the last 20 timesteps, but do not explicitly tell any of the models that this is a new environment, as we want to see their ability to implicitly detect and generalize to this change. We then ask what the agent will do next from the latest timestep. The qualitative difference is that the underlying behavioral program remains the same, but the spatial context and coordinates defining the agent's path are altered. Models must implicitly recognize that the agent's intent has persisted despite a complete change in the geometry of the world (the state coordinates and wall layout).
> - We thank the reviewer for giving us the opportunity to clarify the program generation process and will include our clarification in the paper to help with reproducibility. When ROTE is given a trajectory at first, we ask the same LLM to generate N candidate programs using the same prompt (which we will include in the code that we will open-source), but maintain a non-zero temperature (we used 1.0 in our experiments) so that there is some natural diversity is what the LLM generates. To improve the diversity further, we utilize “rejuvenation” during the SMC process: if a program’s likelihood is below a threshold after the inference process, we ask the LLM to resample a new program (this process repeats at most once per candidate, but in our code we set this as a free-parameter). During rejuvenation, the LLM takes the original prompt used to generate the initial set of programs, as well as the code for the program that received a low probability, and is asked to create a new agent program which explains the observed data but differs from the low likelihood candidate it is replacing. As shown in Figure 12, this rejuvenation process proves more useful than maintaining the initial set of LLM generated programs when there are fewer hypotheses, however, at larger values of N this effect washes out.

---

> > ### Author Response · Authors · 2025-11-24
> >
> > - We thank the reviewer for the insightful comment about how our algorithm’s impact could be improved by extending to belief inference as well, and have added following paragraph to an extended discussion in our paper: “Lastly, unlike traditional Theory of Mind approaches that predict beliefs and goals, our work focuses solely on action prediction. If we view beliefs as dispositions to act \citep{Ramsey1927-RAMVFA, Ryle1949-RYLTCO-7}, predicting a distribution over an agent's internal decision-making states and logic for transitioning between them is functionally equivalent to belief inference. ROTE is designed to excel in scenarios dominated by predictable, routine, or script-like behaviors, such as daily routines in warehouses and stores, relatively stable social conventions like driving, or routine household settings. This is because ROTE exploits the efficiency of executing simple code for long-horizon prediction in these routine settings. For ROTE to gain true generality and address the rigidity concern, future work is explicitly focused on extending it to generate Probabilistic Programming Languages (PPLs), such as \textit{memo}, which is specialized for social reasoning in JAX \citep{chandra2025memo}. This extension would allow ROTE to infer the distribution over actions or latent mental states, directly addressing the stochastic nature of human actions without abandoning the executable code format. In terms of failure modes, domains requiring high-fidelity continuous control over raw sensor data (e.g., video feeds) require ROTE's inferred high-level programs to be integrated into a Task and Motion Planning architecture, where ROTE provides the symbolic task plan to a low-level neural control mechanism. Finally, for deeply complex, goal-directed behaviors involving ``unknown unknowns'' in partially observable environments, the very notion of a fixed FSM-like programmatic model may be fundamentally unsuitable, indicating that in these cases, the representation itself is too rigid to capture the agent's full intentionality. Thus, we view ROTE as generating and reasoning over one of many possible representations that are suitable for behavior prediction rather than a catch-all.”

---

> > > ### Comment · Reviewer_vNqa · 2025-11-27
> > >
> > > Thanks for the clarifications. I still feel that focusing mainly on script-like action prediction is not enough to call it modeling others’ minds. At the moment, it feels more like “modeling others’ script-like behaviors as code.” Without representing a broader range of mental states such as beliefs, some important aspects of human minds are missing. However, I appreciate the idea and contribution of the paper, and therefore I maintain my original rating.

---

### Official Review · Reviewer_5dXd · 2025-11-02

**Soundness:** 3
**Presentation:** 3
**Contribution:** 3
**Rating:** 6
**Confidence:** 4

**Summary:**

This paper engages with the problem of social interaction modeling, proposing to model agents' predictable patterns as behavioral programs. The authors introduce ROTE, an algorithm that uses LLMs to synthesize behavioral programs and probabilistic inference for reasoning about uncertainty. By evaluating on gridworld tasks and embodied simulators, ROTE predicts human and AI behaviors well and outperforms baselines.

**Strengths:**

- This paper represents an interesting and novel idea. Using program synthesis with SMC inference for modeling agentic behavior is worth studying.
- This paper shows good empirical results: ROTE outperforms baseline conditions. It also demonstrates superior generalization accuracy.
- This paper has a reasonable selection of environment. It goes beyond simple 2D gridworld (Construction) with the inclusion of an embodied environment (Partnr).
- This paper also condutcs human studies, allowing the reader to have a clearer sense of the performance of ROTE. That fact that ROTE achieves human-level performance while the baselines do not here is noteworthy.
- The paper is generally well written.

**Weaknesses:**

- The paper has Naive LLM (NLLM) as a baseline, which seems to be the strongest baseline. But why not also use Chain-of-Thought (few or zero shot) as a baseline? How would that compare to ROTE? (Or could you argue that CoT is not appropriate here?) I consider this to be a major weakness of the paper, and would raise my score if this issue is addressed.
- Using programs to represent behavioral patterns can be a good idea, but it might be too rigid or inflexible in many cases. I would appreciate that the authors discuss this aspect more. For example, the first limitation at the end is related to this. But I think more discussions would be helpful. For example, as things currently stand, what kinds of real-world domains or settings do the authors expect ROTE to mostly work? For domains where ROTE may not work, would it be because the programs are not expressive enough (e.g., wrt high-dimentional inputs) or programs as a presentation is not suitable?

**Questions:**

See "Weaknesses".

---

> ### Author Response · Authors · 2025-11-24
>
> - We appreciate the reviewer’s question about how ROTE might compare to the Chain-of-Thought baselines. Following the reviewer’s suggestions, we ran the zero-shot CoT baseline across a number of open-source models (Llama-3.1-8b-Instruct, DeepseekV2-Lite, Deepseek-V2-Coder-Lite-Instruct, Qwen2.5-3b-Instruct, Qwen2.5-7b-Instruct, Qwen2.5-14b-Instruct). We used zero-shot because the full trajectory of (state, action) pairs takes a significant amount of context, limiting the number of examples we can add. For a fair comparison, all of our results with ROTE also were done “zero-shot” without examples of successful programs beyond the code skeleton it should fill in. Interestingly, we found that in this setup CoT did not improve compared to ROTE and NLLM, even though most of its prompts and hyperparameters were the same as NLLM. The best performing model (Qwen2.5-14B-Instruct) for CoT was 26.91 % accurate in Construction, 11.50 % accurate on Human data, and 42% accurate on Partner. The same model used in ROTE had an accuracy of 63.78% in Construction, 50% on Human data, and 50% on Partnr. We will be sure to include these results in the paper (Tables 1, 2, and 3), as well as a full accuracy breakdown by model, a sample of our prompts, and evaluations on larger open models and more model families.
>
> - We thank the reviewer for the insightful point regarding the potential inflexibility of programmatic representations for modeling complex behavior. We agree that ROTE's core assumption of modeling agents as code is not ideal for every situation. We will edit the discussion section to have the following paragraph: “ROTE is designed to excel in scenarios dominated by predictable, routine, or script-like behaviors, such as daily routines in warehouses and stores, relatively stable social conventions like driving, or routine household settings. This is because ROTE exploits the efficiency of executing simple code for long-horizon prediction in these routine settings. For ROTE to gain true generality and address the rigidity concern, future work is explicitly focused on extending it to generate Probabilistic Programming Languages (PPLs), such as [memo](https://github.com/kach/memo), which is specialized for social reasoning in JAX. This extension would allow ROTE to infer the distribution over actions or latent mental states, directly addressing the stochastic nature of human actions without abandoning the executable code format. In terms of failure modes, domains requiring high-fidelity continuous control over raw sensor data (e.g., video feeds) require ROTE's inferred high-level programs to be integrated into a Task and Motion Planning architecture, where ROTE provides the symbolic task plan to a low-level neural control mechanism. Finally, for deeply complex, goal-directed behaviors involving "unknown unknowns" in partially observable environments, the very notion of a fixed FSM-like programmatic model may be fundamentally unsuitable, indicating that in these cases, the representation itself is too rigid to capture the agent's full intentionality. Thus, we view ROTE as generating and reasoning over one of many possible representations that are suitable for behavior prediction rather than a catch-all.”

---

### Meta-Review · Area_Chair_YvBX · 2026-01-11

**Summary:**

This paper proposes ROTE, a framework for modeling others’ behavior by representing recurring, predictable patterns as executable programs rather than policies over beliefs and desires. The core idea is to treat action understanding as a program synthesis problem: large language models are used to generate candidate behavioral programs, and probabilistic inference is applied to maintain uncertainty over these hypotheses and predict future actions. The approach is evaluated across gridworld environments and a large-scale embodied household simulator, demonstrating strong performance in both single-step and multi-step prediction settings, including improved generalization from sparse observations and competitive results relative to behavior cloning and LLM-based baselines.

Overall, reviewers viewed this as a technically interesting and well-motivated contribution that sits between inverse planning and purely data-driven approaches, with the idea of modeling humans using “scripts” also rooted in the social science literature. The strengths highlighted include the framing of behavior prediction as executable code, the integration of LLM-based hypothesis generation with Bayesian inference, and a fairly extensive empirical evaluation across multiple environments and agent types. The main concerns centered on comparisons with prior work and stronger models (e.g., chain-of-thought prompting), as well as the discussion of limitations of the proposed approach (e.g., rigidity and computational cost).

**Reviewer Concerns:**

The main concerns centered on comparisons with prior work and stronger models (e.g., chain-of-thought prompting), as well as the discussion of limitations of the proposed approach, including rigidity of the programmatic representation and computational cost.

Several reviewers questioned whether the experimental evaluation sufficiently positioned ROTE relative to closely related prior work and to stronger LLM-based baselines. In particular, they asked whether methods such as chain-of-thought prompting or larger, more capable models could achieve comparable performance without the additional modeling machinery. In response, the authors added new comparisons and experiments in the rebuttal, which strengthen the empirical case and mitigate concerns that the reported gains are driven by weak baselines. The authors also provide discussion on several of the prior works listed by the reviewers.

Reviewers also raised concerns about the limitations of modeling behavior as executable programs, especially with respect to flexibility and computational overhead. Questions were raised about how well the approach would extend to more complex or stochastic behaviors, and about the cost of the initial program synthesis stage relative to simpler baselines. The authors clarify the intended scope of the approach and provide additional discussion and measurements addressing these trade-offs, framing these issues as limitations of applicability rather than fundamental weaknesses of the method.

One reviewer also raised concerns about the definition of scripted behavior, the degree of structural constraints in the program space, and the computational cost of program synthesis. The authors provided clarifications and additional analysis addressing these points.

**Reviewer Scores:**

Two of the reviewers were initially positive about the work, with scores of 6.

Reviewer P8y6 gave an initial score of 4, with the main concerns focused on the positioning of this work in the literature. The authors responded with comparisons to and discussion of the prior work listed by the reviewer. Reviewer G1C7 also gave an initial score of 4, with concerns regarding the discussion of the literature, the models, and the evaluations, including several concrete questions. The authors provided reasonably detailed responses to these points. Overall, I would expect at least one of the initial scores of 4 to be raised to 6 based on the rebuttal.

Overall, this is a borderline paper. I have read the paper myself, and I find the main idea of modeling human interactions using “scripts” to be interesting and grounded in the social science literature. With the expectation that the authors will incorporate their responses into the final version, I believe this paper would make an interesting contribution to the field and therefore recommend acceptance.

---

### Decision · Program_Chairs · 2026-01-26

Accept (Poster)